# The CNK–HYP scaffolding complex promotes RAF activation by enhancing KSR–MEK interaction

Pierre Maisonneuve [1,2,8] ✉, Malha Sahmi[3,8], Fanny Bergeron-Labrecque[3], Xianjie Iris Ma [2], Juliette Queguiner[3], Geneviève Arseneault[3], Martin Lefrançois[3], Igor Kurinov[4], Rémi Fronzes [5], Frank Sicheri [2,6] ✉ & Marc Therrien [3,7] ✉

The RAS–MAPK pathway regulates cell proliferation, differentiation and survival, and its dysregulation is associated with cancer development. The pathway minimally comprises the small GTPase RAS and the kinases RAF, MEK and ERK. Activation of RAF by RAS is notoriously intricate and remains only partially understood. There are three RAF isoforms in mammals (ARAF, BRAF and CRAF) and two related pseudokinases (KSR1 and KSR2). RAS-mediated activation of RAF depends on an allosteric mechanism driven by the dimerization of its kinase domain. Recent work on human RAFs showed that MEK binding to KSR1 promotes KSR1–BRAF heterodimerization, which leads to the phosphorylation of free MEK molecules by BRAF. Similar findings were made with the single *Drosophila* RAF homolog. Here we show that the fly scaffold proteins CNK and HYP stabilize the KSR–MEK interaction, which in turn enhances RAF–KSR heterodimerization and RAF activation. The cryogenic electron microscopy structure of the minimal KSR–MEK–CNK–HYP complex reveals a ring-like arrangement of the CNK–HYP complex allowing CNK to simultaneously engage KSR and MEK, thus stabilizing the binary interaction. Together, these results illuminate how CNK contributes to RAF activation by stimulating the allosteric function of KSR and highlight the diversity of mechanisms impacting RAF dimerization as well as the regulatory potential of the KSR–MEK interaction.

Given its evolutionary conservation in metazoans, simpler biological models such as *Drosophila* and *Caenorhabditis elegans* have been extensively employed to identify new components of the RAS–ERK pathway and to characterize complex regulatory mechanisms[1]. For example, genetic screens in flies and nematodes identified the pseudokinase KSR, a close RAF homolog, as a key component acting between RAS and RAF[2–4]. The characterization of KSR revealed that its kinase domain heterodimerizes with the kinase domain of RAF, through a surface that

[1]Université de Bordeaux, CNRS, Bordeaux INP, CBMN, UMR 5248, IECB, F-33600, Pessac, France. [2]Center for Molecular, Cell and Systems Biology, Lunenfeld-Tanenbaum Research Institute, Sinai Health System, Toronto, Ontario, Canada. [3]Institute for Research in Immunology and Cancer, Laboratory of Intracellular Signaling, Université de Montréal, Montréal, Quebec, Canada. [4]Department of Chemistry and Chemical Biology, Cornell University, NE-CAT, Argonne, IL, USA. [5]Institut Européen de Chimie et Biologie, Université de Bordeaux-CNRS (UMR 5234), Pessac, France. [6]Departments of Molecular Genetics and Biochemistry, University of Toronto, Toronto, Ontario, Canada. [7]Département de pathologie et biologie cellulaire, Université de Montréal, Montréal, Quebec, Canada. [8]These authors contributed equally: Pierre Maisonneuve, Malha Sahmi. ✉e-mail: p.maisonneuve@iecb.u-bordeaux.fr; sicheri@lunenfeld.ca; marc.therrien@umontreal.ca

**Fig. 1 | Crystal structure of a CNK–HYP complex. a**, Domain architecture of *Drosophila* CNK and HYP. Amino acid positions are indicated on top of each depiction. Boundaries of constructs used in this study are represented as solid lines underneath. **b**, S2 cells were transfected with the indicated plasmid constructs. GST pull-downs were conducted on total cell lysates. Antibodies used to detect protein levels in the GST pull-downs or total cell lysates are indicated to the right of each blot. Co-expressed Pyo-HYP, V5-KSR and Myc-MEK proteins associate with GST-CNK²⁻⁵⁴⁹ and GST-CNK²⁻³⁸⁴, but not with GST-CNK²⁻²⁰⁹, which lacks the PDZ domain. Results shown here are representative of at least three

independent experiments. Signal quantifications relative to control lanes 1–3 (set at 1.00) are shown under the blots. *$P \leq 0.05$ using a one-way ANOVA test. Exact $P$ values for quantified signals are as follows. KSR levels: lane 4, $P = 0.9775$; lane 6, $P = 0.9999$; lane 7, $P < 0.0001$; lane 8, $P < 0.0001$; lane 9, $P = 0.4961$. MEK levels: lane 5, $P > 0.9999$; lane 6, $P = 0.0018$; lane 7, $P < 0.0001$; lane 8, $P < 0.0001$; lane 9, $P = 0.891$. HYP levels: lane 4, $P = 0.9611$; lane 5, $P = 0.9997$; lane 7, $P = 0.0012$; lane 8, $P = 0.0007$; lane 9, $P > 0.9999$. **c**, Ribbon and surface representations of the CNK²⁻³³⁰–HYP complex obtained by X-ray crystallography.

is conserved across all RAF family members, to allosterically activate RAF kinase activity[5]. Subsequent studies demonstrated that kinase domain dimerization is an essential step for the activation of all catalytically competent RAF family members, including ARAF, BRAF and CRAF in mammals[6,7]. Furthermore, recent work with mammalian and fly proteins demonstrated that MEK binding to KSR selectively drives RAF–KSR heterodimerization and RAF activation, thus identifying KSR proteins as MEK-dependent allosteric activators of RAF[8].

Similar to KSR, the protein Connector eNhancer of KSR (CNK) was identified by genetic means in *Drosophila* as a key regulatory component of the RAS–ERK pathway[9]. CNK is a scaffolding protein comprising an amino-terminal conserved sterile alpha motif (SAM),

conserved region in CNK (CRIC) and PSD-95/Dlg/ZO-1 (PDZ) domains, followed by a central pleckstrin homology (PH) domain[10]. In flies, CNK, like KSR, acts at a step between RAS and RAF to facilitate RAF activation, but its precise mechanism of action remains unresolved. Notably, the N-terminal part of CNK (denoted CNK^NT) encompassing the SAM, CRIC and PDZ domains and a 260-residue linker preceding the PH domain can recruit the KSR–MEK complex in a manner dependent on its physical association with the SAM-domain-containing protein hyphen (HYP; also known as Aveugle)[11,12]. The interaction of CNK and HYP is mediated in part by a canonical head-to-tail interaction of their SAM domains[13]. Whether the interaction of CNK–HYP with KSR–MEK is direct, and the specific role of domains beyond the SAM domain of CNK in RAS–ERK

pathway regulation, remain open questions. Here, we functionally and structurally characterized the KSR–MEK–CNK–HYP complex to reveal the mechanism by which KSR–MEK binds CNK–HYP and how this contributes to RAS–ERK signaling.

## Results

### Structure of CNK^NT bound to HYP

In previous experiments, the minimal fragment of CNK interacting with the KSR–MEK complex in the presence of HYP extended from amino acids 2 to 549 (CNK$^{2-549}$; Fig. 1a)[11]. These experiments did not address whether KSR and/or MEK could independently associate with CNK$^{2-549}$. To address this point, we separately expressed a GST-CNK$^{2-549}$ fusion construct along with the different pairwise combinations of Pyo-tagged HYP, V5-tagged KSR and Myc-tagged MEK constructs in S2 cells and assessed their ability to associate following GST pull-downs. Interestingly, although basal interactions with CNK$^{2-549}$ were observed for HYP, KSR and MEK when tested alone or in pairs (Fig. 1b, lanes 1–6), co-expression of HYP, KSR and MEK greatly enhanced their association with CNK$^{2-549}$ (lane 7). These observations suggest cooperative interaction between CNK, HYP, KSR and MEK.

To identify the minimal portion of CNK able to associate with the KSR–MEK complex, we probed the requirement of domains in addition to the SAM domain of CNK. To this end, we generated two additional carboxy-terminal deletions fused N-terminally to GST; namely, CNK$^{2-384}$ lacking most of the linker region between the PDZ and PH domains, and CNK$^{2-209}$ that additionally lacks the PDZ domain (Fig. 1a). As assessed using GST pull-downs, co-expression of epitope-tagged HYP, KSR and MEK proteins also interacted strongly with the CNK$^{2-384}$ truncation (lane 8). By contrast, the CNK$^{2-209}$ truncation was severely impaired in its ability to associate with all three proteins (Fig. 1b, lane 9). Together, these results indicate that the formation of a tight complex between CNK, HYP, KSR and MEK is dependent on sequences in addition to the SAM domains; in particular, the PDZ domain.

Given that the CNK$^{SAM}$–HYP$^{SAM}$ domain interaction is necessary but not sufficient for KSR–MEK recruitment, we sought structural information on CNK beyond the SAM domain using CNK$^{2-330}$, which minimally comprises the SAM, CRIC and PDZ domains. We determined a 2.1 Å resolution structure of the complex of CNK$^{2-330}$ with HYP$^{2-106}$ by single anomalous dispersion phasing using selenomethionine substituted proteins (see Extended Data Fig. 1 and Table 1 for X-ray diffraction and refinement statistics). The CNK–HYP complex adopts a rigid four-lobe torus (or ring) with extensive interactions between all four domains (Fig. 1c). As expected, the HYP$^{SAM}$ domain interacts with the CNK$^{SAM}$ domain in a prototypical head-to-tail manner with a buried contact surface of 518 Å$^2$ described previously (Extended Data Fig. 1b)[13]. The atypically long helix α5 in HYP$^{SAM}$ facilitates a non-canonical interaction with the CNK$^{CRIC}$ domain (Fig. 1c), which itself adopts a four helical bundle resembling the FAT domain of the FAK kinase family[14] (Extended Data Fig. 1c). A buried contact surface of 961 Å$^2$ between HYP$^{SAM}$ and CNK$^{CRIC}$ is composed predominantly of hydrophobic residues on helix α5 of HYP$^{SAM}$ and on helices α2 and α3 of CNK$^{CRIC}$ (Extended Data Fig. 2a). The PDZ domain adopts a canonical fold (Extended Data Fig. 1d) and, despite its C-terminal position within the SAM-CRIC-PDZ sequence, adopts a central position between the CNK$^{SAM}$ and CNK$^{CRIC}$ domains in the toroid structure, making no direct contacts with HYP$^{SAM}$ (Fig. 1c). Furthermore, none of the inter-domain interactions mediated by the PDZ domain are canonical in nature[15]. A buried contact surface between CNK$^{PDZ}$ and CNK$^{SAM}$ of 667 Å$^2$ comprises a mixture of hydrophobic and hydrophilic interactions (Extended Data Fig. 2d). A buried contact surface between CNK$^{PDZ}$ and CNK$^{CRIC}$ of 296 Å$^2$ is mediated largely by hydrophilic interactions (Extended Data Fig. 2g).

Next, we sought to validate the functional relevance of the CNK–HYP structure by testing the impact of specific point mutations on the binding of CNK to HYP in cells and on the ability of CNK and HYP to induce RAS–ERK pathway signaling as evidenced by the

## Table 1 | Data collection and refinement statistics

| | SeMet crystal[a] | Native crystal (PDB 8BW8)[b] |
|---|---|---|
| **Data collection** | | |
| Space group | P 31 | P 31 |
| Cell dimensions | | |
| a, b, c (Å) | 131.7, 131.7, 71.4 | 131.9, 131.9, 71.5 |
| α, β, γ (°) | 90, 90, 120 | 90, 90, 120 |
| Resolution (Å) | 114.0–3.46 (3.79–3.46)[c] | 114.2–2.1 (2.14–2.10) |
| $R_{merge}$ | 0.133 (0.241) | 0.062 (1.66) |
| I / σI | 31.9 (20.5) | 14.0 (1.0) |
| Completeness (%) | 100.0 (100.0) | 99.80 (100.0) |
| Redundancy | 44.3 (42.9) | 5.2 (5.4) |
| **Refinement** | | |
| Resolution (Å) | | 2.1 |
| No. of reflections | | 81,051 (4,475) |
| $R_{work}$ / $R_{free}$ | | 0.187/0.216 |
| No. of atoms | | |
| Protein | | 8,416 |
| Ligand/ion | | 54 |
| Water | | 283 |
| B-factors | | |
| Protein | | 64.63 |
| Ligand/ion | | 91.29 |
| Water | | 59.52 |
| R.m.s. deviations | | |
| Bond lengths (Å) | | 0.002 |
| Bond angles (°) | | 0.54 |

[a]Statistics for the SeMet crystals correspond to 16 merged datasets. [b]Statistics for the Native crystal correspond to 1 dataset. [c]Values in parentheses are for the highest-resolution shell.

phosphorylation of MEK. With respect to the CNK$^{CRIC}$–HYP$^{SAM}$ interface (Extended Data Fig. 2a), point mutations targeting hydrophobic interactions between helix α3 of CNK$^{CRIC}$ (I123D, L152D and I123D_L152D) and helix α5 of HYP (L90D, I97D and L90D_I97D) significantly reduced the association between Flag-CNK$^{2-549}$ and GST-HYP as assessed by GST pull-down (Extended Data Fig. 2b). Previous work showed that the phosphorylation of MEK in S2 cells is greatly induced by the CNK–HYP complex in the presence of co-expressed RAS$^{V12}$, RAF and KSR[13]. Although single-site HYP and CNK mutations had little to no adverse effects on the induction of pMEK levels under these conditions, the double-site mutations in either CNK$^{CRIC}$ or HYP$^{helix α5}$ strongly impaired phospho-MEK (pMEK) levels (Extended Data Fig. 2c), thus validating the functional relevance of the affected binding interfaces. Interestingly, the presence of overexpressed RAS$^{V12}$, RAF, KSR and MEK mitigated, to various degrees, the impact of the single point mutations on the CNK–HYP interaction (compare Extended Data Fig. 2b and 2c). However, this effect was not seen with the double mutants, which probably explains their greater impact on pathway signaling (that is, pMEK levels) (Extended Data Fig. 2c). This observation is consistent with the noted stabilizing influence of the KSR–MEK complex on the CNK–HYP interaction (Fig. 1b).

With respect to the CNK$^{SAM}$–CNK$^{PDZ}$ interface (Extended Data Fig. 2d), mutagenesis of contact residues in each domain, singly or in combination, prevented CNK–HYP association in most instances

(Extended Data Fig. 2e). The same mutations were tested for their impact on CNK–HYP-induced pMEK levels, but most had no effects (Extended Data Fig. 2f). Moreover, the effect of these mutations on the CNK–HYP interaction were also offset by the presence of RAS[V12], RAF, KSR and MEK proteins (Extended Data Fig. 2f). Nevertheless, two sets of double mutations (CNK[PDZ_E250R_V252R] and CNK[SAM_F74R]_CNK[PDZ_E250R]) greatly affected HYP binding in the presence of the other RAS pathway components and considerably reduced pMEK levels (Extended Data Fig. 2f). These results confirm the importance of the CNK[SAM]–CNK[PDZ] contact surface for complex formation and signaling activity.

Finally, with respect to the CNK[CRIC]–CNK[PDZ] interface (Extended Data Fig. 2g), we introduced mutations to disrupt a salt interaction between D81 and K283. An individual mutation of each residue to the opposite charge (D81R and K283D) reduced complex formation partially, while the combined mutation restored CNK–HYP interaction to wild-type levels (Extended Data Fig. 2h). This result is consistent with the notion that a salt interaction between positions 81 and 283 (with either polarity) contributes favorably to CNK–HYP structural integrity. We also assessed the impact of these mutations on CNK–HYP-induced MEK phosphorylation. Only the K283D single-site mutation weakly but reproducibly reduced pMEK levels (Extended Data Fig. 2i). Consistent with this result, the K283D mutation also hampered complex formation in the presence of the RAS pathway components (Extended Data Fig. 2i). Importantly, this defect, as well as the reduction in pMEK levels, were fully rescued by co-introduction of the D81R mutation (Extended Data Fig. 2i), which restores a favorable salt interaction but with opposite polarity. These results support the relevance of the CNK[CRIC]–CNK[PDZ] interface and provide further evidence for the validity of the toroid CNK[SAM-CRIC-PDZ]–HYP[SAM] crystal structure.

## CNK–HYP binds directly to KSR–MEK

Our structural and functional characterization of the CNK–HYP complex delineated the minimal portion of CNK required for HYP binding and for KSR–MEK recruitment. We previously showed that an isolated kinase domain of KSR bound to MEK is also efficiently recruited to CNK–HYP in cells[15]. However, it is unclear whether this interaction is direct. To address this issue, we separately produced and purified the CNK[2–330]–HYP and KSR[654–966]–MEK complexes to homogeneity using bacterial and insect cell (Sf9) expression systems, respectively and then determined their ability to form a higher-order complex upon mixing. In gel filtration chromatography, the CNK–HYP and KSR–MEK complexes exhibited distinct elution profiles (Extended Data Fig. 3a,e). Markedly, when the two complexes were pre-incubated together, a significant proportion of the CNK–HYP complex shifted to higher molecular weight fractions that overlapped with the KSR–MEK complex (Extended Data Fig. 3a). These results indicate that CNK, HYP, KSR and MEK proteins can associate directly to form a higher-order complex.

## Mapping interaction surfaces on CNK and KSR

We next sought to map the surfaces on CNK–HYP that mediate KSR–MEK binding. We reasoned that if the interaction were functionally relevant, then the contact residues would be evolutionarily conserved. Projection of conserved residues onto the surface of the CNK–HYP structure identified multiple residues as candidates for mutagenesis, with the majority on the same surface of the toroid structure (Fig. 2a and Supplementary Figs. 1 and 2). Of ten sites mutagenized on HYP and CNK, a single mutation (M197E) within the linker joining the CNK[CRIC] and CNK[PDZ] domains reproducibly reduced the binding of CNK–HYP to KSR, as determined by GST pull-downs (Fig. 2b,c). Follow-on mutagenesis of adjacent conserved CNK residues, including D195K and V198E (but not Q200A), also disrupted the interaction of CNK–HYP with KSR–MEK (Fig. 2b,d). Importantly, none of these mutations altered the interaction of CNK with HYP (Fig. 2e). As an orthogonal approach, we tested a subset of purified mutant proteins for interaction by gel filtration chromatography. Although the D195K, M197E and V198E mutations in CNK did not impair the elution profile of the CNK–HYP complex on their own, they hampered the ability of CNK–HYP to assemble with KSR–MEK (Extended Data Fig. 3b–d). Together, these results identify a conserved surface in the vicinity of the CRIC-PDZ linker of CNK that mediates a direct interaction with KSR–MEK.

To ascertain the functional relevance of the predicted binding surface on CNK for KSR–MEK, we tested mutations targeting this area for impact on CNK–HYP-induced pMEK levels in the presence of RAS[V12], RAF, KSR and MEK. As shown in Fig. 2f, the three strongest mutations in CNK[2–549] affecting complex formation with KSR–MEK (D195K, M197E, V198E) significantly decreased pMEK levels.

We next asked whether expression of CNK mutants with impaired interaction with KSR–MEK would adversely affect phospho-MAPK (pMAPK) levels induced by an activated RTK after depletion of endogenous CNK by RNAi. To this end, we introduced the D195K_V198E double mutation into a full-length CNK construct lacking its natural 3′UTR. Expression of a constitutively active sevenless RTK (SEV[S11] (ref. 16))

**Fig. 2 | The CRIC-PDZ linker region of CNK is required for recruiting KSR–MEK. a**, Ribbon and surface representations of the CNK[2–330]–HYP complex. Projection of evolutionarily conserved residues is shown as a grayscale (black being the most conserved) onto the surface of the CNK–HYP structure. The positions of conserved residues targeted for mutagenesis are indicated and color-coded according to their domain location. **b**, Ribbon representation of the CNK[CRIC-PDZ] linker region. Side chains of residues targeted for mutagenesis are depicted as sticks. **c–g**, S2 cells were transfected with the indicated plasmid constructs. Antibodies used to detect protein levels are indicated to the right of each blot. GST pull-downs in **c–f** were conducted on total cell lysates for each condition. The M197E mutation targeting the CNK[CRIC-PDZ] linker region reduces KSR–MEK recruitment to the CNK–HYP complex (**c**); the D195K, V198E and M197E mutations in the CNK[CRIC-PDZ] linker region greatly impede the ability of CNK–HYP to recruit KSR–MEK (**d**); mutagenesis of the CNK[CRIC-PDZ] linker region does not impinge on the formation of the binary CNK–HYP interaction (**e**), but impairs CNK–HYP's ability to promote RAF activation as assessed by pMEK levels following co-expression of Ha-RAS[V12], Pyo-RAF, V5-KSR and Myc-MEK (RRKM) (**f**). **g**, S2 cells were incubated with dsRNA targeting the 3′UTR of endogenous *cnk* transcripts 24 h before transfection. SEV[S11] expression was induced by 30 min heat shock at 37 °C and incubated for 2 h before collecting the cells. The ability of CNK to promote SEV[S11]-induced MAPK activation is impeded by the double D195K_V198E mutation targeting the CNK[CRIC-PDZ] linker region as assessed by pMAPK levels. Experiments in **c–g** were repeated at least three times. Signal quantifications relative to control lanes (set at 1.00) are shown under relevant blots. *$P \le 0.05$ using a one-way ANOVA test. Exact $P$ values for quantified signals in **c** are as follows. KSR levels: lane 3, $P = 0.9994$; lane 4, $P = 0.9992$; lane 5, $P = 0.9968$; lane 6, $P = 0.9994$; lane 7, $P = 0.0761$; lane 8, $P = 0.9891$; lane 9, $P = 0.9821$; lane 10, $P = 0.9868$; lane 11, $P = 0.9385$; lane 12, $P = 0.9995$; lane 13, $P = 0.9968$. CNK levels: lane 3, $P = 0.8486$; lane 4, $P = 0.6118$; lane 5, $P = 0.6821$; lane 6, $P = 0.858$; lane 7, $P = 0.1317$; lane 8, $P = 0.9996$; lane 9, $P = 0.1848$; lane 10, $P = 0.8816$; lane 11, $P = 0.4895$; lane 12, $P = 0.9872$; lane 13, $P = 0.9877$. Exact $P$ values for quantified signals in **d** are as follows. KSR levels: lane 3, $P = 0.9946$; lane 4, $P < 0.0001$; lane 5, $P < 0.0001$; lane 6, $P < 0.0001$; lane 7, $P = 0.0009$; lane 8, $P < 0.0001$. MEK levels: lane 3, $P = 0.9478$; lane 4, $P < 0.0001$; lane 5, $P = 0.0001$; lane 6, $P < 0.0001$; lane 7, $P = 0.037$; lane 8, $P < 0.0001$. CNK levels: lane 3, $P = 0.9937$; lane 4, $P = 0.0003$; lane 5, $P = 0.0182$; lane 6, $P = 0.0003$; lane 7, $P = 0.8326$; lane 8, $P = 0.0004$. Exact $P$ values for quantified signals in **e** are as follows. CNK levels: lane 3, $P = 0.7654$; lane 4, $P = 0.5781$; lane 5, $P = 0.7979$; lane 6, $P = 0.6404$; lane 7, $P = 0.8454$; lane 8, $P = 0.9997$. Exact $P$ values for quantified signals in **f** are as follows. KSR levels: lane 4, $P > 0.9999$; lane 5, $P = 0.0004$; lane 6, $P = 0.0066$; lane 7, $P = 0.0004$; lane 8, $P = 0.4077$. MEK levels: lane 4, $P = 0.9997$; lane 5, $P < 0.0001$; lane 6, $P < 0.0001$; lane 7, $P < 0.0001$; lane 8, $P = 0.0036$. CNK levels: lane 4, $P = 0.7734$; lane 5, $P = 0.0006$; lane 6, $P = 0.0019$; lane 7, $P = 0.0005$; lane 8, $P = 0.2225$. pMEK levels: lane 4, $P = 0.2198$; lane 5, $P = 0.0081$; lane 6, $P = 0.4627$; lane 7, $P = 0.0138$; lane 8, $P = 0.9997$. Exact $P$ values for quantified signals in **g** are as follows. pMAPK levels: lane 3, $P = 0.0017$; lane 4, $P = 0.3245$; lane 5, $P = 0.0011$.

under a heat-shock promoter in S2 cells induces pMAPK levels (Fig. 2g). Co-transfection of a double-stranded RNA (dsRNA) targeting the 3'UTR of *cnk* transcripts reduced pMAPK levels as expected (Fig. 2g, lane 3) and

this effect was fully restored by co-expressing an RNAi-insensitive CNK wild-type construct (lane 4). By contrast, the D195K_V198E mutation in CNK failed to rescue pMAPK levels under these conditions (lane 5).

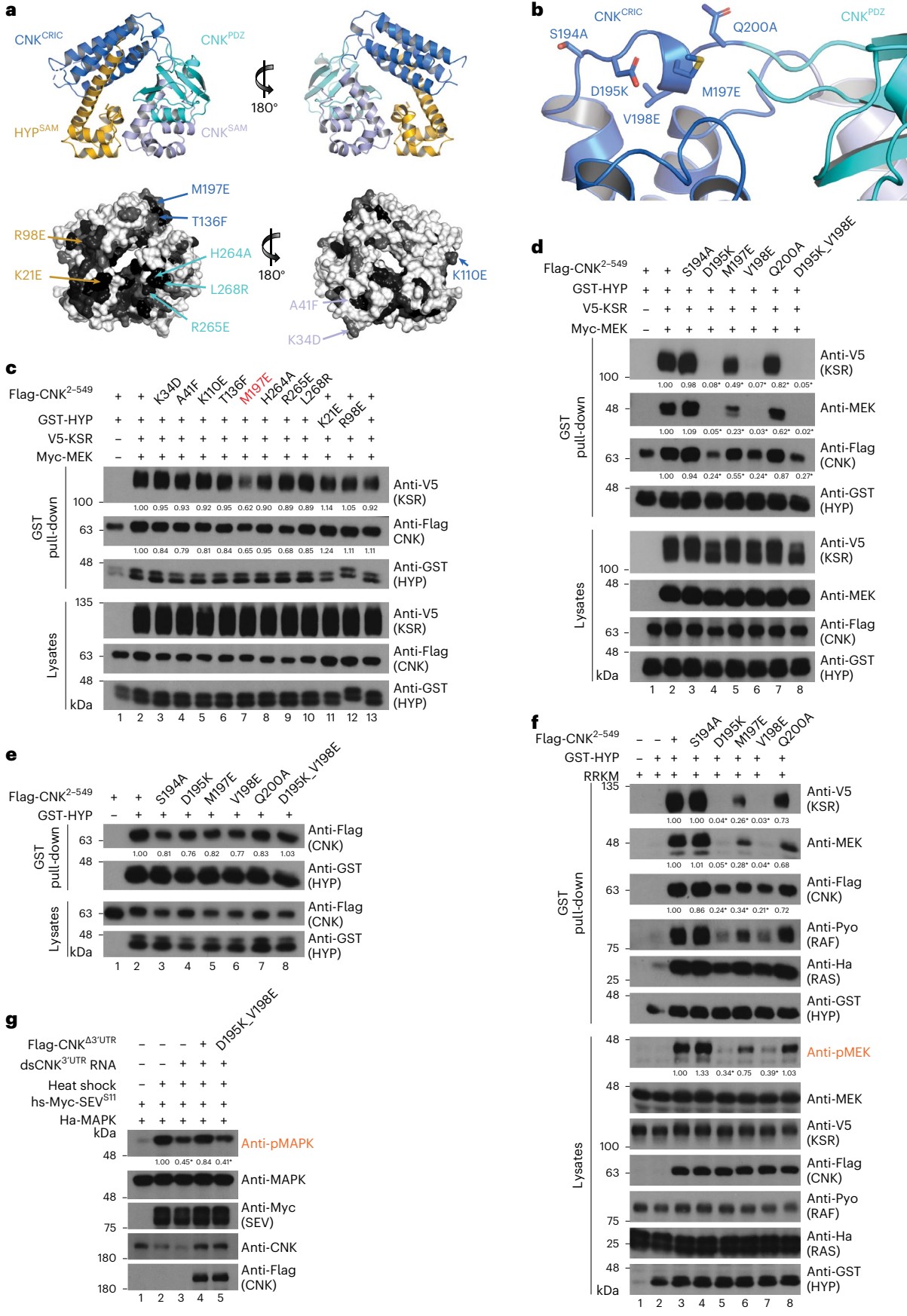

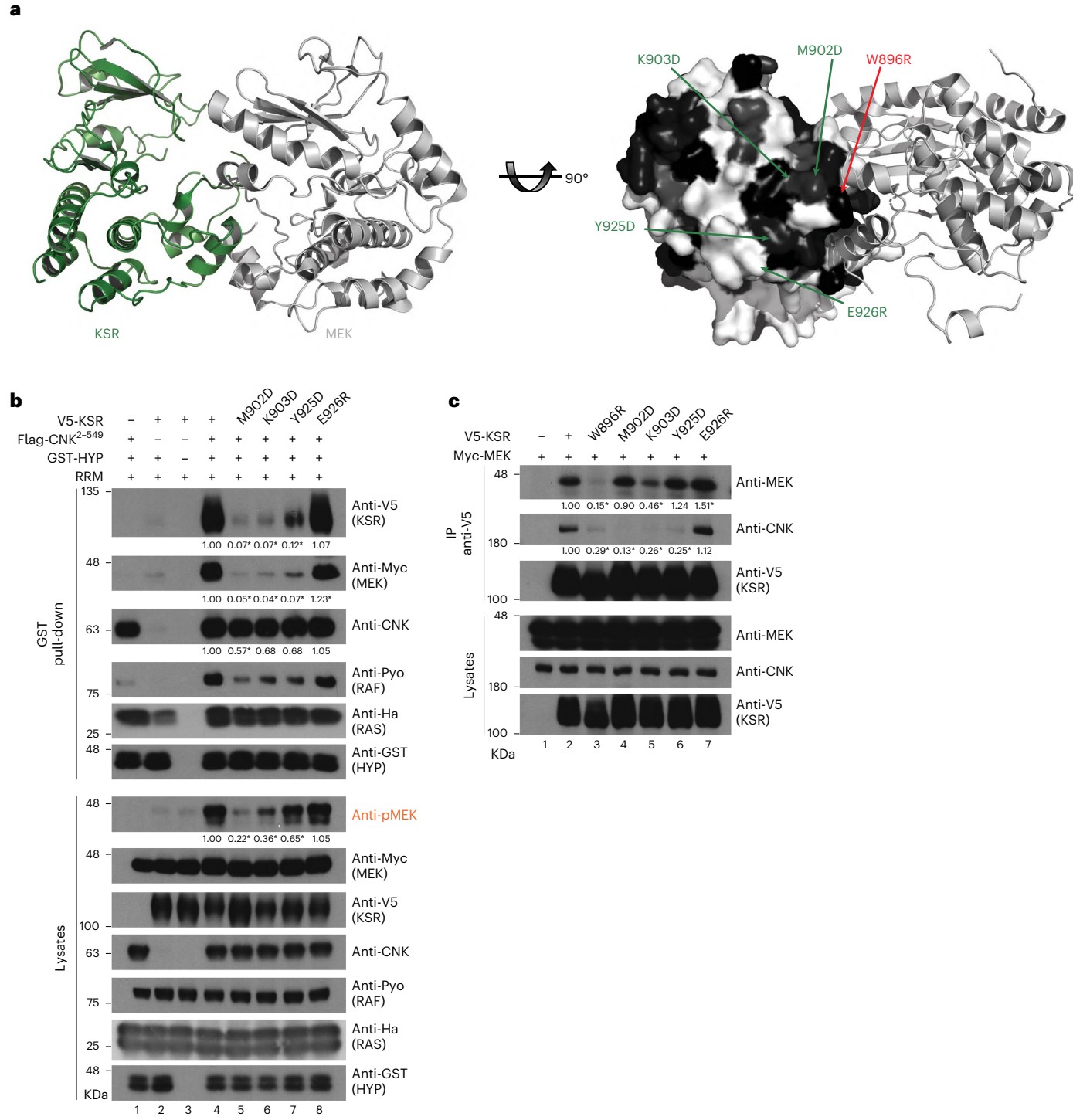

**Fig. 3 | Identification of surface residues on KSR enabling interaction with CNK–HYP. a**, Ribbon representations of a face-to-face KSR2–MEK1 kinase domain dimer (PDB 2Y4I (ref. 25)). Projection of evolutionarily conserved residues (shown in grayscale, with black being most conserved) onto the surface of the KSR structure. Residues targeted for mutagenesis are indicated by arrows. The red-colored mutation targets a residue within the αG helix of KSR (W896) that is a known contact point with MEK[8]. **b,c**, S2 cells were transfected with the indicated plasmid constructs. Antibodies used to detect protein levels are indicated to the right of each blot. GST pull-downs (**b**) or anti-V5 immunoprecipitations (**c**) were conducted on total cell lysates. **b**, Mutagenesis of conserved residues located at the base of the C-lobe of KSR prevents interaction with the CNK–HYP complex and impedes RAF activation. The mutation of a non-conserved residue in close vicinity (E926R) has no effect. RRM denotes co-transfection of Ha-RAS$^{V12}$, Pyo-RAF

and Myc-MEK. **c**, Mutations of the conserved residues in **b** (with the exception of W896A) have no or a marginal effect on the binary KSR–MEK interaction. Experiments in **b** and **c** were repeated at least three times. Signal quantifications relative to control lanes (set at 1.00) are shown under relevant blots. *$P \le 0.05$ using a one-way ANOVA test. Exact $P$ values for quantified signals in **b** are as follows. KSR levels: lane 5, $P < 0.0001$; lane 6, $P < 0.0001$; lane 7, $P < 0.0001$; lane 8, $P = 0.702$. MEK levels: lane 5, $P < 0.0001$; lane 6, $P < 0.0001$; lane 7, $P < 0.0001$; lane 8, $P < 0.0001$. CNK levels: lane 5, $P = 0.0153$; lane 6, $P = 0.0718$; lane 7, $P = 0.0773$; lane 8, $P = 0.9763$. pMEK levels: lane 5, $P < 0.0001$; lane 6, $P < 0.0001$; lane 7, $P < 0.0001$; lane 8, $P = 0.697$. Exact $P$ values for quantified signals in **c** are as follows. MEK levels: lane 3, $P = 0.0015$; lane 4, $P = 0.9683$; lane 5, $P = 0.0335$; lane 6, $P = 0.5215$; lane 7, $P = 0.0421$. CNK levels: lane 3, $P = 0.0338$; lane 4, $P = 0.0096$; lane 5, $P = 0.026$; lane 6, $P = 0.0255$; lane 7, $P = 0.973$.

Together, these results provide compelling evidence that a surface involving the CRIC–PDZ linker region of CNK plays an important role in RTK-driven RAS–MAPK signaling in *Drosophila* cells by mediating direct interaction with KSR–MEK.

After defining the probable binding surface on CNK for KSR–MEK, we next sought to identify the complementary interaction surface on the KSR–MEK complex. Projection of conserved residues onto the surface of the known KSR2–MEK1 crystal structure (PDB 7JUR (ref. 17)) identified candidate regions for mutagenesis on KSR (Fig. 3a and Supplementary Fig. 3). We previously found that mutations in KSR that impaired MEK binding also affected CNK binding[11], suggesting that the binding surface on KSR could be close to MEK. Therefore, we first examined three conserved residues (M902, K903 and Y925) located at the base of the C-lobe of KSR, adjacent to the C-lobe of MEK. Notably, the three single-site mutations M902D, K903D and Y925D strongly reduced binding of KSR–MEK to CNK–HYP and also impaired pMEK induction to varying degrees (Fig. 3b). By contrast, a negative control mutation (E926R) had no effect.

Given that the three conserved residues in KSR that are relevant for CNK–HYP binding are close to the MEK binding surface, we next tested whether their mutations affected binding of KSR to MEK (that is, to rule out an indirect mechanism of action). Compared with the αG helix mutation W896R in KSR, which is known to abrogate MEK binding[8], the M902D and Y925D mutations had no adverse effect on KSR binding to MEK, whereas the K903D mutation had an intermediate effect (Fig. 3c). As an orthogonal approach, we tested the KSR^M902D and KSR^Y925D mutations for their effect on KSR–MEK interaction with CNK–HYP by gel filtration chromatography. Both mutations on their own or combined perturbed the interaction of KSR–MEK with CNK–HYP (Extended Data Fig. 3e–h). Together, these results identify a surface on the C-lobe of KSR centered on M902 and Y925 as the probable direct binding site for CNK–HYP.

## Cryogenic electron microscopy structure of KSR–MEK–CNK–HYP

To visualize the binding mode of KSR–MEK to CNK–HYP, we pursued an atomic structure of the minimal KSR–MEK–CNK–HYP complex in the presence of AMPPNP and the MEK inhibitor trametinib by cryogenic electron microscopy (cryo-EM). The resultant structure, determined to 3.3 Å resolution (Fig. 4a, Table 2 and Extended Data Fig. 4), revealed unambiguous Coulomb potential density for the entire CNK–HYP subcomplex, all of MEK including bound AMPPNP and trametinib with the exception of helix α1 (Extended Data Fig. 5a,b), and all of KSR kinase domain with the exception of the N-lobe (Extended Data Fig. 5c). Disorder of the N-lobe of KSR correlated with the absence of ligand in the kinase active site. Aside from the apparent disorder of the KSR N-lobe, the binding mode of KSR to MEK is similar to that observed for the isolated KSR–MEK structure[17], with a small 3° pivot of the KSR C-lobe with respect to MEK that eliminates potential for steric clash with CNK^CRIC (Extended Data Fig. 6). The activation segments of KSR and MEK adopt similar inactive-like conformations stabilized by an

intermolecular antiparallel β-sheet as previously observed[17,18]. We note, however, that the activation segment of KSR in the KSR–MEK–CNK–HYP structure adopts a more unstructured conformation in comparison to the isolated KSR–MEK structure, which may be because of the disordered nature of the N-lobe of KSR in the former structure (Extended Data Fig. 6d).

The binding surface on CNK–HYP for KSR–MEK is composed of all three domains of CNK (SAM, CRIC and PDZ), whereas HYP makes no direct contact with KSR–MEK (Fig. 4a). The largest contact is made between CNK and KSR (765 Å$^2$ buried surface area) and involves the αG–αH linker and helix αH of KSR and helices α1 and α4 of CNK^CRIC and the CNK^CRIC–CNK^PDZ linker (Extended Data Fig. 7a,b). The second largest contact is made between CNK^CRIC and MEK (719 Å$^2$ buried surface area) and involves helices α1 and α2 of CNK^CRIC and the αG–αH and the αH–αI linkers of MEK (Extended Data Fig. 7c). The third largest contact is made between CNK^SAM and MEK (330 Å$^2$ buried surface area) and involves the α3–α4 linker, helix α4 and helix α5 of CNK^SAM and the APE motif-αF linker, αH–αI linker and helix αI of MEK (Extended Data Fig. 7d). Lastly, the C-terminal extension of the PDZ domain makes minor contact with the C-lobe of the KSR kinase domain (216 Å$^2$ buried surface area; Extended Data Fig. 5d).

Overall, the CNK–HYP complex binds across the bottom surfaces of both KSR and MEK kinase C-lobes. Consistent with our contact mapping studies (Figs. 2 and 3), the interaction perturbing D195K, M197E and V198E mutations in CNK and the M902D, K903D and Y925D mutations in KSR directly map to the interface between CNK–HYP and KSR–MEK (Extended Data Fig. 7a). To further validate the KSR–MEK–CNK–HYP structure, we sought to mutagenize residues at additional points of contact between CNK–HYP and KSR–MEK. As above, the mutants were tested for their ability to disrupt binding of CNK–HYP to KSR–MEK (while preserving the binary interactions of CNK with HYP and of MEK with KSR) and for the ability to perturb pathway signaling (Extended Data Fig. 7). Although most single mutations (KSR R913E, MEK E323R, CNK^SAM R46F and Y76R, CNK^CRIC N96E and R99E) had little to no effect on quaternary complex formation or pathway signaling, three mutations (CNK^CRIC N82E, N82W and Q84E) disrupted the interaction between CNK–HYP and KSR–MEK, while the CNK^CRIC N82W mutation additionally abolished pathway signaling (Extended Data Fig. 7e,f). Given that binding of CNK–HYP to KSR–MEK appeared resistant to single mutations in earlier studies (Extended Data Fig. 2), we tested the effect of the new mutations in combination. Strikingly, all tested combinations strongly disrupted the interaction between CNK–HYP and KSR–MEK and reduced MEK phosphorylation (Extended Data Fig. 7g,h) without affecting the binary interactions of KSR with MEK or CNK with HYP (Extended Data Fig. 7i,j). Together, these results validate the functional relevance of the cryo-EM structure of CNK–HYP bound to KSR–MEK.

## Comparison with 14-3-3 bound RAF complexes

The structure of KSR determined here corresponds to an isolated kinase domain, which raises the question of how other domains and

**Fig. 4 | Cryo-EM structure of the KSR–MEK–CNK–HYP complex.**
**a**, Cryo-EM map (left) and ribbon representations (right) of the KSR–MEK–CNK–HYP complex. CNK contacts the C-lobes of KSR and MEK kinase domains. KSR engages the CRIC domain of CNK, while MEK engages both the SAM and CRIC domains of CNK (see Extended Data Figs. 6 and 7 for details). **b–e**, S2 cells were transfected with the indicated plasmid constructs. Antibodies used to detect protein levels are indicated to the right of each blot. Anti-V5 (**b**) and anti-Pyo (**c**) immunoprecipitations were conducted on total cell lysates. The CNK–HYP complex enhances KSR–MEK complex formation (**b**), which drives RAF–KSR heterodimerization and RAF transactivation (**c**). Mutations at the binding interface between CNK and KSR disrupt the ability of CNK–HYP to drive RAF–KSR heterodimerization and RAF transactivation. The ability of CNK–HYP to promote RAF activity depends on the canonical side-to-side interface of the

RAF–KSR heterodimer (**d**) and on the BRS-CC-SAM interaction between the RAF and KSR N-terminal regulatory regions[8] (**e**). **f**, Model depicting the role of the CNK–HYP complex in KSR–MEK-driven RAF activation. Experiments in **b–e** were repeated at least three times. Signal quantifications relative to control lanes (set at 1.00) are shown under relevant blots. *$P \leq 0.05$ using a one-way ANOVA test. Exact $P$ values for quantified signals in **b** are as follows. MEK levels: lane 3, $P = 0.0432$. Exact $P$ values for quantified signals in **c** are as follows. KSR levels: lane 3, $P = 0.7991$; lane 4, $P = 0.0125$; lane 5, $P = 0.869$; lane 6, $P = 0.36$. pMEK levels: lane 4, $P = 0.0042$; lane 5, $P = 0.5287$; lane 6, $P = 0.9031$. Exact $P$ values for quantified signals in **d** are as follows. pMEK levels: lane 3, $P = 0.9977$; lane 4, $P = 0.9065$; lane 5, $P < 0.0001$; lane 6, $P = 0.9976$; lane 7, $P = 0.9784$. Exact $P$ values for quantified signals in **e** are as follows. pMEK levels: lane 3, $P < 0.0001$; lane 4, $P < 0.0001$; lane 5, $P = 0.0608$.

binding partners of KSR might relate to and/or influence binding to CNK–HYP. In quiescent cells, a 14-3-3 dimer stabilizes BRAF in a monomeric auto-inhibited state through simultaneous binding of the 14-3-3 dimer to a pair of phospho-sites that N-terminally and C-terminally flank the kinase domain of BRAF[19,20]. Additional stabilizing interactions are afforded by the CRD domain of BRAF and by the binding

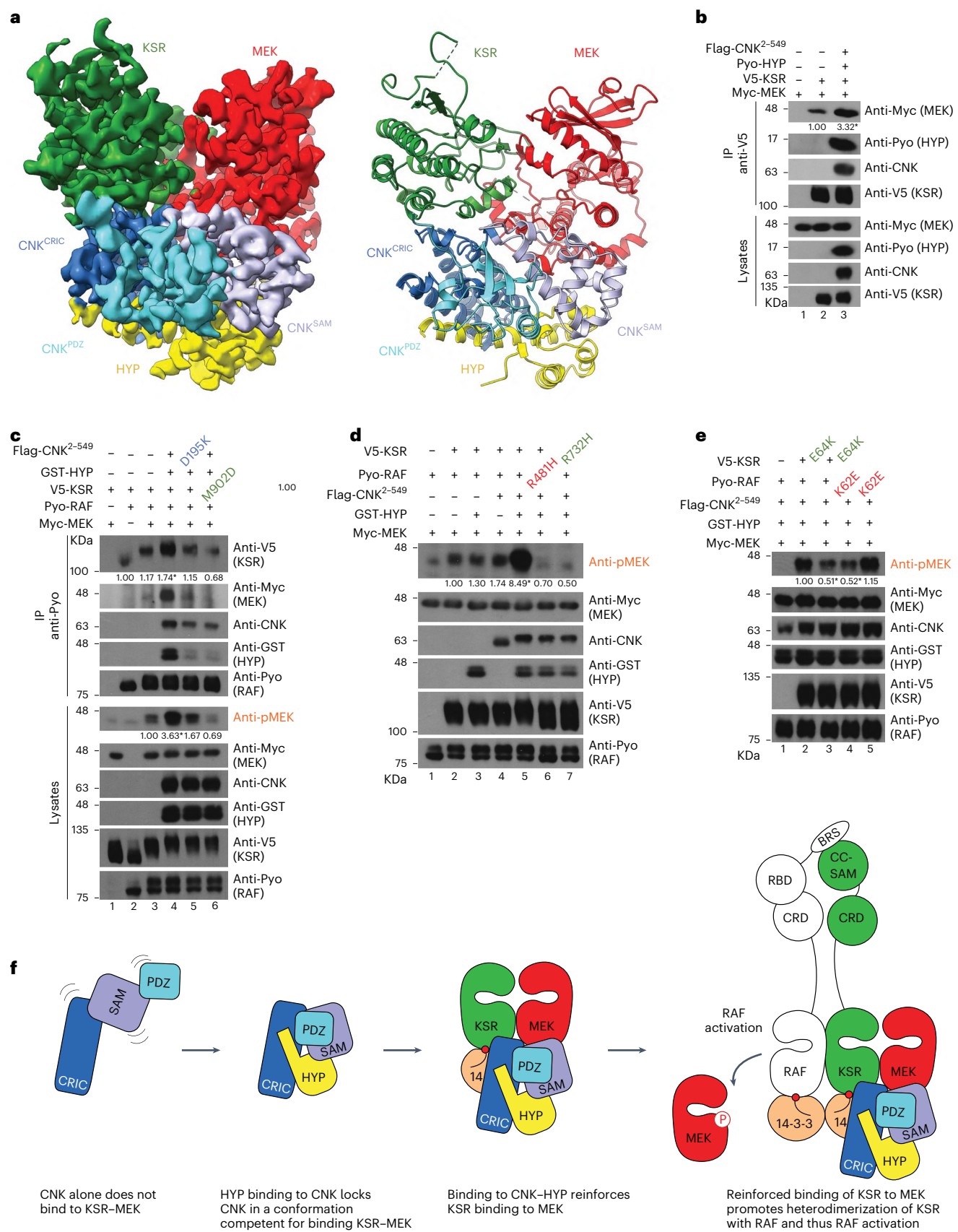

of BRAF to MEK (Extended Data Fig. 8a, left)[19,20]. As the two 14-3-3 phospho-regulatory sites, the kinase domain and the CRD domain of BRAF are conserved in KSR proteins[21], we reasoned that KSR could in principle adopt a similar auto-inhibited conformation. Superposition of the auto-inhibited structure of BRAF with the cryo-EM structure of KSR–MEK–CNK–HYP reveals a small steric clash between CNK[PDZ] and 14-3-3 bound to the N-terminal phospho-site in BRAF (Extended Data Fig. 8a, right). A 14-3-3 dimer also stabilizes the active state of BRAF by binding to the C-terminal phospho-regulatory sites of two separate BRAF molecules, thereby orienting the RAF kinase domains in a productive side-to-side dimer configuration (Extended Data Fig. 8b, left)[19,20,22]. As the C-terminal regulatory sites and the side-to-side kinase domain dimerization surfaces of BRAF are conserved in KSR proteins, KSR probably adopts a similar active state structure when bound to RAF. Superposition of the KSR–MEK–CNK–HYP structure with the 14-3-3 promoted BRAF active-state structure revealed no steric clashes. (Extended Data Fig. 8b, right). Although CNK–HYP binding to KSR appears compatible with KSR modeled in both 14-3-3 promoted active-dimer and inactive-monomer states, in the absence of subtle conformational changes to accommodate minor steric clashes, we reason that CNK–HYP binding may favor the former.

## CNK–HYP activates RAF by stabilizing KSR–MEK

Our cryo-EM structure reveals an extensive interaction of CNK–HYP with both KSR and MEK kinase domains, suggesting that CNK–HYP stabilizes the interaction between KSR and MEK. We previously showed that MEK binding to KSR1 leads to BRAF allosteric activation by promoting the dimerization of KSR1 and BRAF kinase domains[8]. This led us to reason that the mechanism by which CNK–HYP activates RAS–ERK pathway signaling is mediated through the same allosteric mechanism.

To investigate this model, we tested the ability of the CNK[2–549]–HYP complex to enhance the association between KSR and MEK in pull-down experiments. Compared to control (Fig. 4b, lane 2), co-expression of CNK[2–549]–HYP in S2 cells strongly increased the ability of MEK to co-immunoprecipitate with KSR. Thus, it appears that the CNK–HYP complex promotes and/or stabilizes the formation of the KSR–MEK complex.

Next, we tested whether CNK–HYP binding to KSR–MEK affects the ability of KSR–MEK to interact with RAF. RAF weakly associates with KSR when co-expressed in S2 cells (Fig. 4c, lane 2) and, as expected, this interaction is enhanced by co-expression of MEK (lane 3). Strikingly, co-expression of the CNK–HYP complex in these conditions strongly enhanced RAF association with KSR, which correlated with an enhancement of pMEK levels (lane 4). Demonstrating the relevance of the direct interaction between CNK–HYP and KSR–MEK complexes in RAF–KSR dimerization, the CNK[D195K] and KSR[M902D] variants, which abrogates CNK–HYP binding to KSR–MEK (Figs. 2 and Fig. 3 and Extended Data Fig. 3), also disrupted the ability of CNK–HYP to promote RAF–KSR interaction and to enhance pMEK levels (Fig. 4c, lanes 5 and 6).

To confirm that RAF–KSR dimerization induced by CNK–HYP is canonical in nature, we examined the effect of RAF[R481H] and KSR[R732H] mutations on the side-to-side dimerization surfaces of the kinase domains, which disable allosteric activation[5]. As shown in Fig. 4d, CNK–HYP did not stimulate MEK phosphorylation with the RAF[R481H] or KSR[R732H] dimer interface mutants (lanes 6 and 7) compared to wild-type proteins (lane 5). Thus, the stimulatory effect of CNK–HYP on MEK phosphorylation depends on the allosteric mechanism by which KSR normally transactivates RAF.

The dimerization of BRAF with KSR1 also involves a direct interaction between the N-terminal regulatory region of both proteins, mediated by the BRS domain in BRAF and the CC-SAM domain in KSR[8]. To determine whether binding of the BRS and CC-SAM domains is also required for pathway activation by CNK–HYP, we introduced charge reversal mutations in two evolutionarily conserved residues (K62E in BRS and E64K in CC-SAM) known to form a salt bridge in the human

## Table 2 | Cryo-EM data collection, refinement and validation statistics

| | KSR–MEK–CNK–HYP (EMD-16281), (PDB 8BW9) |
|---|---|
| **Data collection and processing** | |
| Magnification | ×45,000 |
| Voltage (kV) | 200 |
| Electron exposure (e⁻/Å²) | 50.53 |
| Defocus range (μm) | −0.5 to −2.6 |
| Pixel size (Å) | 0.93 |
| Symmetry imposed | C1 |
| Initial particle images (no.) | 2,451,838 |
| Final particle images (no.) | 141,531 |
| Map resolution (Å) | 3.32 (0.143) |
| FSC threshold | |
| Map resolution range (Å) | 2.97–48.73 |
| **Refinement** | |
| Initial model used (PDB code) | 7JUR, 8BW8 |
| Model resolution masked/unmasked (Å) | 3.29/3.34 (0.143) |
| FSC threshold | |
| Model resolution range (Å) | 2.97–7.78 |
| Map sharpening B factor (Å²) | −90.34 |
| Model composition | |
| Non-hydrogen atoms | 6,957 |
| Protein residues | 861 |
| Ligands | 2 |
| B factors (Å²) | |
| Protein | 119.48 |
| Ligand | 150.01 |
| R.m.s. deviations | |
| Bond lengths (Å) | 0.003 |
| Bond angles (°) | 0.578 |
| **Validation** | |
| MolProbity score | 1.62 |
| Clashscore | 6.56 |
| Poor rotamers (%) | 0.0 |
| Ramachandran plot | |
| Favored (%) | 96.21 |
| Allowed (%) | 3.79 |
| Disallowed (%) | 0.0 |

BRAF–KSR1 dimer[8]. As shown in Fig. 4e, both mutations impaired CNK–HYP-induced pMEK levels. However, co-expression of the two mutant proteins fully restored pMEK levels, probably caused by the restoration of a functional salt bridge with opposite polarity between these two residues (lane 5). Together, these findings confirm that the CNK–HYP complex stimulates RAS–ERK signaling by promoting the KSR–MEK interaction, which in turn induces RAF–KSR dimerization and transactivation.

## Discussion

RAS-mediated RAF activation is a multi-step process initiated by the interaction of GTP-loaded RAS with the RAS-binding domain of RAF at

the plasma membrane[21]. Relocalization of RAF to the plasma membrane is followed by dephosphorylation of the 14-3-3 N-terminal phospho-site of RAF that precedes the kinase domain. This releases inhibitory binding of one protomer in a 14-3-3 protein dimer while leaving the C-terminal phospho-site interaction intact. The kinase domain of RAF then undergoes dimerization with itself or another member of the RAF family (including the KSR pseudokinases), stabilized by a 14-3-3 dimer binding to the C-terminal phospho-sites of the two kinases. Together, these events allosterically induce the catalytic function of RAF[21]. Our study reveals a new twist to the RAF activation mechanism by showing how a scaffolding complex composed of two proteins, CNK and HYP, contributes to RAF activation. RAF activation in flies is highly dependent on its dimerization with the pseudokinase KSR[8], and recent work has demonstrated how this event relies on the interaction between KSR and MEK (the substrate of RAF)[8]. Here, we show that CNK in the presence of HYP binds directly to KSR and MEK. The absence of a direct binding interaction between HYP and KSR–MEK suggests that HYP acts by rigidifying CNK in a conformation competent for KSR–MEK binding (Fig. 4f). The binding of CNK–HYP simultaneously to KSR and MEK in turn reinforces the KSR–MEK interaction, which stimulates the formation of RAF–KSR heterodimers and, ultimately, the allosteric activation of RAF (Fig. 4f). Given that RAF can exist as both symmetric homodimers and asymmetric heterodimers with KSR, the ability of CNK–HYP to drive KSR heterodimerization with RAF has the potential to influence the equilibrium between dimer configurations to fine-tune the activation of the RAS–ERK pathway. Overall, our work shows that the KSR–MEK interaction can serve as an effective point of control for RAS–ERK pathway regulation, which was previously not thought to be the case[23].

Interestingly, our structure shows that the CNK–HYP complex binds to the same region of KSR and MEK corresponding to the conserved helices αG and αH of the protein kinase fold (Supplementary Fig. 4). This region is close to the region that mediates substrate recruitment (in particular, helix αG) by KSR, MEK[18], BRAF[24] and also by other protein kinases such as the RNA-dependent protein kinase PKR[25] and the cyclic AMP-dependent protein kinase PKA[26]. This observation further highlights the importance of the αG–αH region as a common element of protein kinase function and control.

Mammalian genomes encode four proteins related to *Drosophila* CNK; namely, CNK1, CNK2, CNK3 and IPCEF1 (ref. 10). Although some of these have been shown to interact with RAF or KSR proteins and modulate RAS–ERK signaling[27–29], their molecular function within the pathway remains largely unexplored. Interestingly, human CNK2 and the HYP homolog SAMD12 have recently been shown to control cancer cell migration by mediating ARF6 activation induced by AXL signaling[30]. This result indicates that minimally, the CNK–HYP complex itself may provide an opportunity for targeted therapeutic intervention. Finally, given that other scaffolding proteins have been found to influence RAS–ERK pathway function (for example, β-arrestin[31], IQGAP1 (ref. 32), paxillin[33]), it will be interesting to determine whether they function in a manner that similarly impinges on RAF dimerization.

## Online content

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

## Methods

### DNA constructs

Copper-inducible pMet constructs were used for most functional and binding studies conducted in S2 cells. pMet constructs used to express Flag-CNK, GST-HYP, V5-KSR, Myc-MEK, Pyo-RAF and Ha-RAS[V12] have been previously described[11,13]. The pBS-pCNK-3×Flag-CNK[Δ3'UTR] used for rescue experiments has been previously described[16]. pMet-GST-CNK[2–549] was generated by cloning a *Drosophila cnk* DNA fragment produced by PCR (encoding residues 2–549) between the Age1 and Not1 sites of the pMet-GST-His-TEV vector[11]. The truncated CNK variants (2–384 and 2–209) were generated by introducing a stop codon at residues 385 and 210, respectively, in pMet-GST-CNK[2–549]. pMet-Pyo-HYP was generated by cloning a PCR product encoding *Drosophila* HYP (positions 2–106) that included three Pyo epitopes (MEYMPME) at its N terminus into the Kpn1 and Not1 sites of the pMet vector. Novel point mutations in CNK, HYP, KSR and RAF constructs were introduced using the QuickChange II Site-Directed mutagenesis kit (Agilent). All constructs, including those described below, were fully verified by Sanger sequencing.

For protein production in bacteria, PCR products encoding wild-type or mutant versions of CNK (2–330) or HYP (2–106) proteins were inserted in tandem in a pGEX-4T backbone. CNK fragments were cloned in-frame at the C terminus of GST-TEV and were followed by 6×His-TEV-HYP sequences that included a synthetic *trc* (*trp-lac*) promoter at the 5' end. The resulting construct, pGEX-4T GST-TEV-CNK[2–330]::trc 6×His-TEV-HYP, allowed the co-expression of CNK and HYP proteins in the same bacterial cells, which enhanced the production of soluble CNK proteins.

We used the pbiGBac baculovirus co-expression system[34] for producing the *Drosophila* complex of KSR kinase domain (KSR[KD]) and MEK in SF9 cells. Two separate library vectors (pLIB) were first generated, namely, pLIB-GST-6×His-TEV-KSR[654–966], which corresponds to the kinase domain of KSR fused to GST at its N terminus, and pLIB-MEK, which encodes full-length MEK isoform B (393 residues). Next, the two gene expression cassettes consisting of the polyhedrin promoter, gene of interest and SV40-terminator sequences were amplified by PCR from the pLIB constructs. Linearized pBIG1a vector and the two PCR products were then assembled in a Gibson assembly reaction as previously described[34] to generate the pBIG1a-GST-6×His-TEV-KSR[654–966]-MEK construct.

### Cell culture, transfections and RNAi

S2 cells (DGRC) were grown in EX-CELL 420 serum-free medium (Sigma-Aldrich) at 27 °C, while heat-inducible Sev[S11]-expressing S2 cells[9] were grown in Schneider's insect medium (Sigma-Aldrich) supplemented with 10% FBS at 27 °C. For transfections, $7 \times 10^6$ S2 cells were seeded in a 60 mm tissue culture dish and transfected the next day with appropriate plasmids using Effectene (Qiagen). dsRNAs were produced and used essentially as previously described[35]. Cells transfected with copper-inducible vectors were induced 24 h post transfection by adding 0.7 mM CuSO$_4$ to the medium, and cells were typically collected 36 h later for analysis. For rescue experiments, $2 \times 10^6$ Sev[S11]-expressing S2 cells were seeded overnight in six-well plates ± dsRNAs (15 µg ml$^{-1}$) targeting *cnk* 3'UTR sequences. Cells were transfected 24 h later with the indicated constructs and cultured for 48 h. Cells were then heat-shocked for 30 min at 37 °C and collected 2 h later.

Oligonucleotide sequences (Integrated DNA Technologies) for the synthesis of *cnk* 3'UTR dsRNAs were:

Top strand
5'-GAATTAATACGACTCACTATAGGGAGAGGATTAGCCCCCGTTTACTTATG-3'
Bottom strand
5'-GAATTAATACGACTCACTATAGGGAGAATTATGTACAGTTGACTTTATTC-3'

### Cell lysates, GST pull-downs, immunoprecipitations and immunoblots

For preparation of cell lysates, cells were gently scraped off dishes with a cell lifter, transferred to 15 ml tubes and spun at 1,000 rpm, 4 °C. Media were then removed by aspiration and cell pellets were resuspended in 500 µl of ice-cold Triton lysis buffer (50 mM Tris at pH 7.5, 150 mM NaCl, 10% glycerol, 0.2% Triton X-100, 1 mM EDTA, 1× phosphatase inhibitor cocktail (Sigma-Aldrich), 1 mM sodium vanadate, 20 µM leupeptin, aprotinin (0.15 U ml$^{-1}$), 1 mM phenylmethylsulfonyl fluoride (PMSF)). Lysing cells were incubated for 15 min in Eppendorf tubes on ice with gentle rocking and then centrifuged at 14,000×*g*, 4 °C for 10 min to remove cell debris.

GST pull-down assays were conducted as previously described[13] except that assays were run overnight at 4 °C and beads were washed four times with 1 ml of ice-cold lysis buffer. Immunoprecipitation assays were conducted by adding antibodies and protein A/G agarose beads (Calbiochem) to cell lysates (~1–2 mg of total proteins), followed by gentle rocking overnight at 4 °C. Beads for both assays were washed three times with ice-cold lysis buffer and resuspended in 100 µl of SDS-loading buffer. Samples were denatured for 5 min in boiling water and resolved on 7%, 10% or 12 % SDS–PAGE. Proteins were transferred to nitrocellulose membranes (Pall Corporation) and blocked with 2% BSA (Sigma-Aldrich) when probed with most primary antibodies, or with 5% SKIM MILK (BioShop) when probed with anti-V5 or anti-pMAPK antibodies. Band intensities were quantified on at least three biological replicates using ImageJ (v. 1.54e)[36]. Statistical analysis was conducted using GraphPad Prism (v. 9.5.1).

Sources and dilutions for primary antibodies were as follows: rabbit polyclonal anti-GST (1:1,000; Cell Signaling, cat. no. 2622), mouse monoclonal anti-Flag, clone M2 (1:5,000; Millipore Sigma-Aldrich, cat. no. F3165), mouse monoclonal anti-V5, clone SV5-Pk1 (1:5,000; Invitrogen, cat. no. 46-0705), rabbit polyclonal anti-MEK1/2 (1:1,000 Cell Signaling Technology, cat. no. 9122), rabbit polyclonal anti-pMEK1/2 S217/221 (1:1,000; Cell Signaling Technology, cat. no. 9121), rabbit monoclonal p44/42MAPK, clone 137F5 (1:1,000; Cell Signaling, cat. no. 4695), mouse monoclonal anti-pMAPK and clone MAPK-YT (1:2,000; Millipore Sigma-Aldrich, cat. no. M8159) antibodies. Mouse monoclonal anti-CNK (1:10; clone 26A6A2), anti-Ha (1:200; clone 12CA5), anti-Myc (1:5; clone 9E10) and anti-Pyo (1:5) antibodies were obtained from the supernatants of hybridoma originally generated in the laboratory of G. M. Rubin[37].

### Protein expression and purification

The construct pGEX-4T GST-TEV-CNK2-330:6×His-TEV-HYP was transformed into BL21-CodonPlus DE3-RIL bacteria (Agilent Technologies) for protein production. Protein expression was performed in TB or M9 minimal media supplemented with L-selenomethionine. Cells were induced at an optical density of 1.5 followed by overnight growth at 18 °C by the addition of IPTG at 1 mM. Collected bacterial pellets were resuspended in lysis buffer composed of 100 mM Bis-Tris pH 6.0, 500 mM NaCl, 25 mM imidazole, 10% glycerol, 5 mM MgCl$_2$, 1 mM TCEP and 1 mM PMSF, and then lysed by homogenization. Lysates were clarified by centrifugation for 40 min at 18,000×*g*. Proteins were bound to nickel-affinity resin (GE Healthcare), eluted with lysis buffer containing 400 mM of imidazole. Proteins eluted from nickel-affinity resin were then bound to glutathione affinity resin (GE Healthcare) and eluted by cleavage of the GST tag with TEV protease. TEV protease was subtracted from the purified CNK–HYP complex using nickel-affinity resin. The CNK–HYP complex was concentrated and then buffer exchanged by size-exclusion chromatography using a Superdex75 24 ml column (GE Healthcare) equilibrated in running buffer (100 mM Bis-Tris pH 6.0, 500 mM NaCl, 10% glycerol, 5 mM MgCl$_2$, 1 mM TCEP). All purification steps were carried out at room temperature (20–25 °C) to avoid protein precipitation.

The GST-6×His-TEV-KSR[654–966] and full-length MEK proteins were co-expressed in Sf9 cells (Thermo Fisher, cat. no. 11496015). Cell pellets

were lysed by sonication and homogenization in 50 mM HEPES pH 7.0, 250 mM NaCl, 1 mM TCEP, benzonase and protease inhibitor cocktails (Roche). After centrifugation, proteins were purified using glutathione resin. Proteins were eluted by cleavage of the GST tag using TEV protease. Proteins were further purified by gel filtration using a 120 ml Superdex 200 column equilibrated with 20 mM HEPES pH 7.0, 500 mM NaCl and 1 mM TCEP. All protein fractions corresponding to greater than 95% purity (as assessed by SDS–PAGE) were pooled, concentrated to 40 mg ml$^{-1}$, and flash-frozen in liquid nitrogen. Protein concentration was determined by ultraviolet–visible absorption spectroscopy at 280 nm wavelength using a NanoDrop spectrophotometer (Thermo Fisher Scientific) and theoretical extinction coefficients. We probed the phosphorylation state of the activation segment of KSR and MEK following gel filtration purification by trypsin digest mass spectrometry. All the detected peptides of the activation segment of KSR correspond to the non-phosphorylated form, suggesting that the activation segment of KSR is not phosphorylated. We detected both phosphorylated and non-phosphorylated forms of peptides corresponding to the activation segment of MEK, indicating that Ser241 (equivalent to S222 in humans) was partially phosphorylated in our insect cell protein preparations.

## Analysis of complex formation by size-exclusion chromatography

The formation of the KSR–MEK–CNK–HYP complex was analyzed by mixing an equimolar ratio of wild-type or mutant forms of the CNK–HYP complex with wild-type or mutant forms of the KSR–MEK complex at a final concentration of 34 µM in a final volume of 100 µl in buffer containing 20 mM HEPES pH 7.0, 500 mM NaCl and 1 mM TCEP. Samples were injected into an S200 HiLoad 10/300 column at 0.75 ml min$^{-1}$ with a fraction size of 1 ml. Fractions were analyzed by SDS–PAGE gel stained by Coomassie blue.

## Protein crystallization and crystal structure determination

CNK–HYP protein complexes were crystallized at 20 mg ml$^{-1}$ at 20 °C in sitting drops with a 1:1 mix of protein and solution containing 0.1 M Bis-Tris propane pH 7.0 and 1.0 M ammonium tartrate dibasic pH 7.0. A native dataset was collected on a flash-frozen crystal cryo-protected in mother liquor containing 20% glycerol at 100 K on the NE-CAT beamline station 24-ID-E, at the Advanced Photon Source at a wavelength of 0.9792 Å. L-selenomethionine-labeled CNK–HYP protein complexes were mixed 50%:50% or 75%:25% with native protein complex and crystallized at 20 mg ml$^{-1}$ at 20 °C in sitting drops with a 1:1 mix of protein and solution containing 0.1 M Bis-Tris propane pH 6.0–6.25 and 1.5–1.3 M ammonium tartrate dibasic pH 7.0. X-ray diffraction was measured on flash-frozen crystals cryo-protected in mother liquor containing 20% glycerol at 100 K on the NE-CAT beamline station 24-ID-C, at the Advanced Photon Source at a wavelength of 0.9791 Å. Data reduction was performed using the XDS package (v. 20170601)[38]. The software BLEND (v. 0.6.23) was then used to select, scale and merge 16 different datasets[39]. Phasing was performed by selenomethionine single-wavelength anomalous dispersion at 5.0 Å using HKL2MAP (v. 0.4b-beta)[40] and the SHELX pipeline (SHELXC v. 2016/1; SHELXD v. 2013/2; SHELXE v. 2018/2)[41]. The atomic model obtained by selenomethionine single-wavelength anomalous dispersion was then used as a search model for molecular replacement with a higher-resolution native dataset using PHASER (v. 2.8.1)[42]. Model building and refinement were performed using COOT (v. 0.9)[43] and PHENIX (v. 1.20.1-4487)[44], respectively. Ramachandran statistics are 97.87% favored, 2.04% allowed and 0.1% outliers. All models were validated using MolProbity (v. 1.20.1-4487)[45]. The data collection and refinement statistics are reported in Table 1.

## Cryo-EM specimen preparation and structure determination

The KSR–MEK–CNK–HYP complex was prepared by mixing 0.48 g l$^{-1}$ (6 µM) KSR–MEK complex with 0.34 g l$^{-1}$ (6.7 µM) CNK–HYP complex to obtain a final concentration of the quaternary complex at 6 µM at

1.1 molar excess of CNK–HYP, supplemented with 1 mM MgCl$_2$, 1 mM AMP–PNP and 50 µM trametinib. The sample was equilibrated at 4 °C for 30 min before centrifugation at 18,000×$g$ for 20 min at 4 °C. The supernatant was spotted on glow-discharged Quantifoil R2/2 Cu grids 200 mesh, and plunge-frozen using a Vitrobot Mark IV (FEI) at 4 °C under 100% humidity. Grids were stored in liquid nitrogen. Data were recorded on a 200 kV Talos Arctica electron microscope (Thermo Fisher Scientific) equipped with Gatan K2 summit direct electron detector. Movies were automatically collected in counting mode using SerialEM (v. 4.0.9)[46] with a pixel size of 0.93 Å. The defocus range was 0.5–2.6 µm and each movie contained 44 frames with a dose per frame of 1.15 e$^-$/Å$^2$ for a total dose of 50.53 e$^-$/Å$^2$.

All cryo-EM data processing was conducted in cryoSPARC (v. v3.2)[47] (Extended Data Fig. 4). Beam-induced full-frame motion was corrected with a built-in cryoSPARC routine, and the contrast transfer functions were estimated using Gctf (v. 1.06)[48]. A total of 1,179,823 blob-picked particles were extracted from 1,652 manually selected movies and corrected for beam-induced local motion. After several rounds of 2D classification, 2D templates from 154,791 selected particles were used for template picking. Then 1,421,783 template-picked particles were extracted from 1,652 manually selected movies and corrected for beam-induced local motion. After several rounds of 2D classification, 2D templates from 83,668 selected particles were used for topaz picking. A total of 1,006,192 topaz-picked particles were extracted and corrected for beam-induced local motion. After several rounds of 2D classification, 208,623 particles were selected and used for 3D ab initio reconstruction and homogenous refinement, yielding an initial map at ~3.8 Å resolution that improved to ~3.3 Å after per-particle CTF refinement and several rounds of 3D classification using hetero-refinement and local refinement using a soft mask encompassing the entire complex.

The crystal structure of the CNK–HYP complex (Fig. 1) and the structure of the KSR–MEK complex from PDB 7JUR[17] were placed into the electron microscopy map. This initial model was manually fitted with UCSF Chimera (v. 1.15)[49]. Model building was performed using COOT (v. 0.9)[43] and refined with PHENIX (v. 1.20.1-4487)[44] and ISOLDE (v. 1.4)[50]. Structural analysis was performed with UCSF Chimera. Figures were made with UCSF Chimera (v. 1.15), UCSF ChimeraX (v. 1.6.1)[51] and PyMol (v. 2.5.3) (www.pymol.org). All cryo-EM data collection and refinement statistics are reported in Table 2.

### Reporting summary

Further information on research design is available in the Nature Portfolio Reporting Summary linked to this article.

## Data availability

All data supporting the findings of the current study are available within the paper and its Supplementary Information or source data files. Coordinates and structures for the CNK–HYP complex and for the KSR–MEK–CNK–HYP complex have been deposited in the Protein Data Bank with accession codes PDB 8BW8 and PDB 8BW9, respectively. The sharpened and associated maps of the KSR–MEK–CNK–HYP complex have been deposited in the Electron Microscopy Data Bank under the accession code EMD-16281. Source data are provided with this paper.

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

## Acknowledgements

We thank the Genomics facility of the Institute for Research in Immunology and Cancer for experimental assistance. Research was supported by Impact Grants from the Canadian Cancer Society (grant nos. 702319 to M.T. and 704116 to F.S.), by grants from the Canadian Institutes of Health Research (grant nos. FDN-388023 to M.T., FDN-143277 and PJT-186218 to F.S and FRN-414829 to P.M.) and by the French Foundation for Medical Research (FRM grant no. AJE202110014546 to P.M.). M.T. and F.S. hold Canada Research Chairs in Intracellular Signaling and in Structural Biology of Signaling. This work is based on research conducted using the Northeastern Collaborative Access Team (NE-CAT) beamlines, which are funded by the National Institute of General Medical Sciences (P30 GM124165). The Eiger 16M detector on 24-ID-E is funded by a National Institutes of Health (NIH) Office of Research Infrastructure Programs (ORIP) High-End Instrumentation (HEI) grant (S10OD021527). This research used resources of the Advanced Photon Source, a United States Department of Energy (DOE) Office of Science User Facility operated for the DOE Office of Science by Argonne National Laboratory under contract no. DE-AC02-06CH11357.

## Author contributions

P.M., M.S., F.S. and M.T. designed the experiments. P.M., M.S., F.S. and M.T. wrote the paper. M.S. and J.Q. conducted co-immunoprecipitation, GST pull-downs, RNAi depletions and transactivation assays. M.S., F.B.-L., G.A. and M.L. generated plasmid constructs. P.M., F.B.-L. and X.I.M. expressed and purified proteins. P.M. and X.I.M. crystallized proteins. P.M. performed size-exclusion chromatography binding assays, crystal data processing, crystal structure determination, grid preparation, cryo-EM data collection, cryo-EM data processing and cryo-EM model reconstruction. I.K. performed crystal data acquisition and assisted with crystal data phasing. R.F. assisted with cryo-EM data collection and cryo-EM data processing.

## Competing interests

F.S. is a founder and consultant of Repare Therapeutics. The other authors declare no competing interests.

## Additional information

**Extended data** is available for this paper at https://doi.org/10.1038/s41594-024-01233-6.

**Correspondence and requests for materials** should be addressed to Pierre Maisonneuve, Frank Sicheri or Marc Therrien.

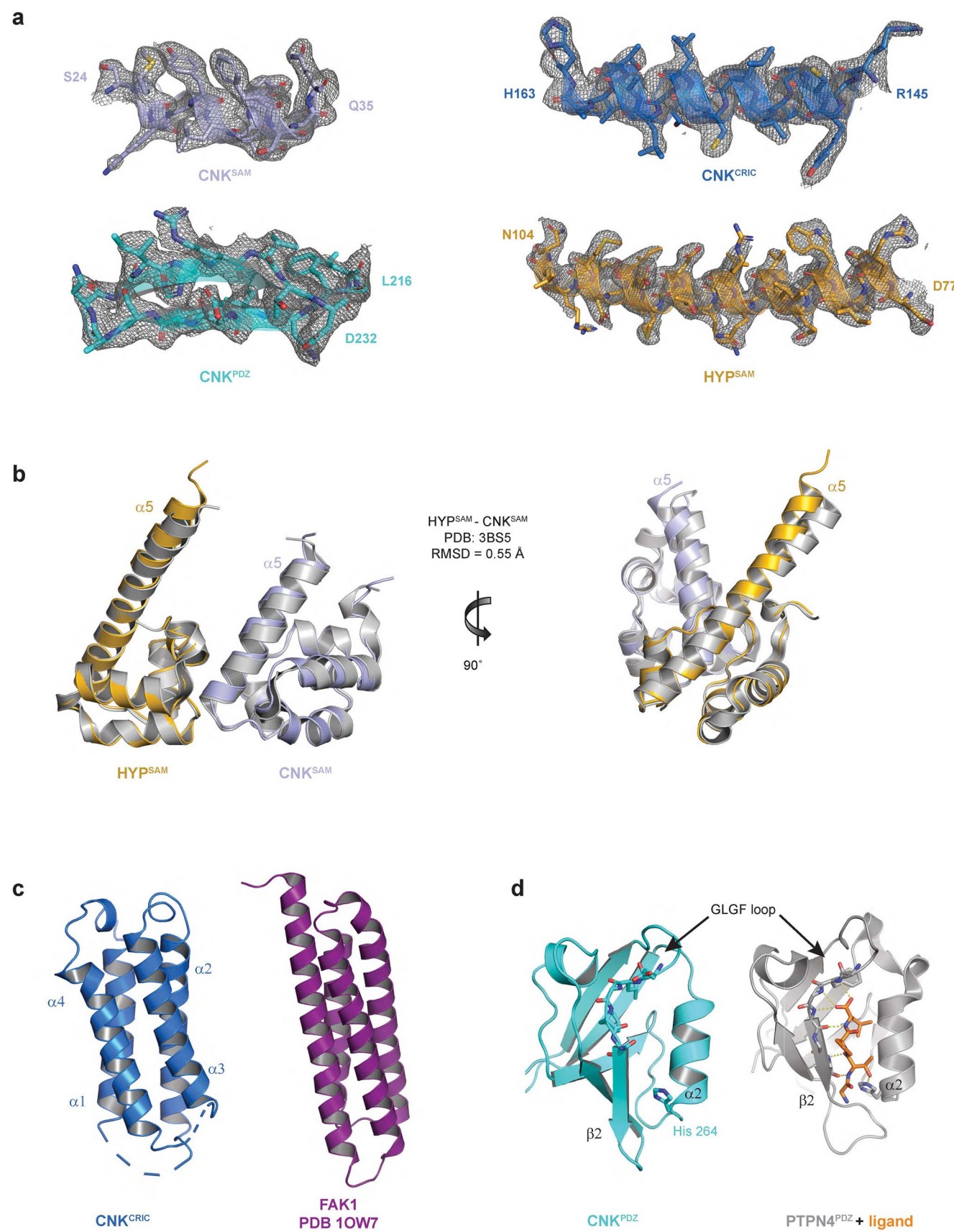

**Extended Data Fig. 1 | See next page for caption.**

**Extended Data Fig. 1 | Crystal structure of the CNK-HYP complex.**
Representative 2Fo−Fc electron density maps contoured at 1.0σ for secondary
structure elements in each domain of the CNK and HYP proteins. **b**, The structure
of the HYP$^{SAM}$-CNK$^{SAM}$ dimer within the CNK-HYP complex is very similar (RMSD
of 0.55 Å) to the structure of the HYP$^{SAM}$-CNK$^{SAM}$ dimer obtained with isolated
SAM domains (PDB ID: 3BS5 (ref. 15)). **c**, The CNK$^{CRIC}$ domain adopts a four-
helix bundle similar in structure (DALI score of 12.6) to the FAT domain of the
Focal Adhesion kinase (PDB ID: 1OW7 (ref. 52)). **d**, The CNK$^{PDZ}$ domain adopts a
canonical fold similar to the PDZ domain in PTPN4. His264 at the N-terminal end
of helix α2 and the conformation of the GLGF loop predict that CNK could bind a
type I PDZ-binding motif as observed for PTPN4 (PDB ID: 5EZ0 (ref. 53)).

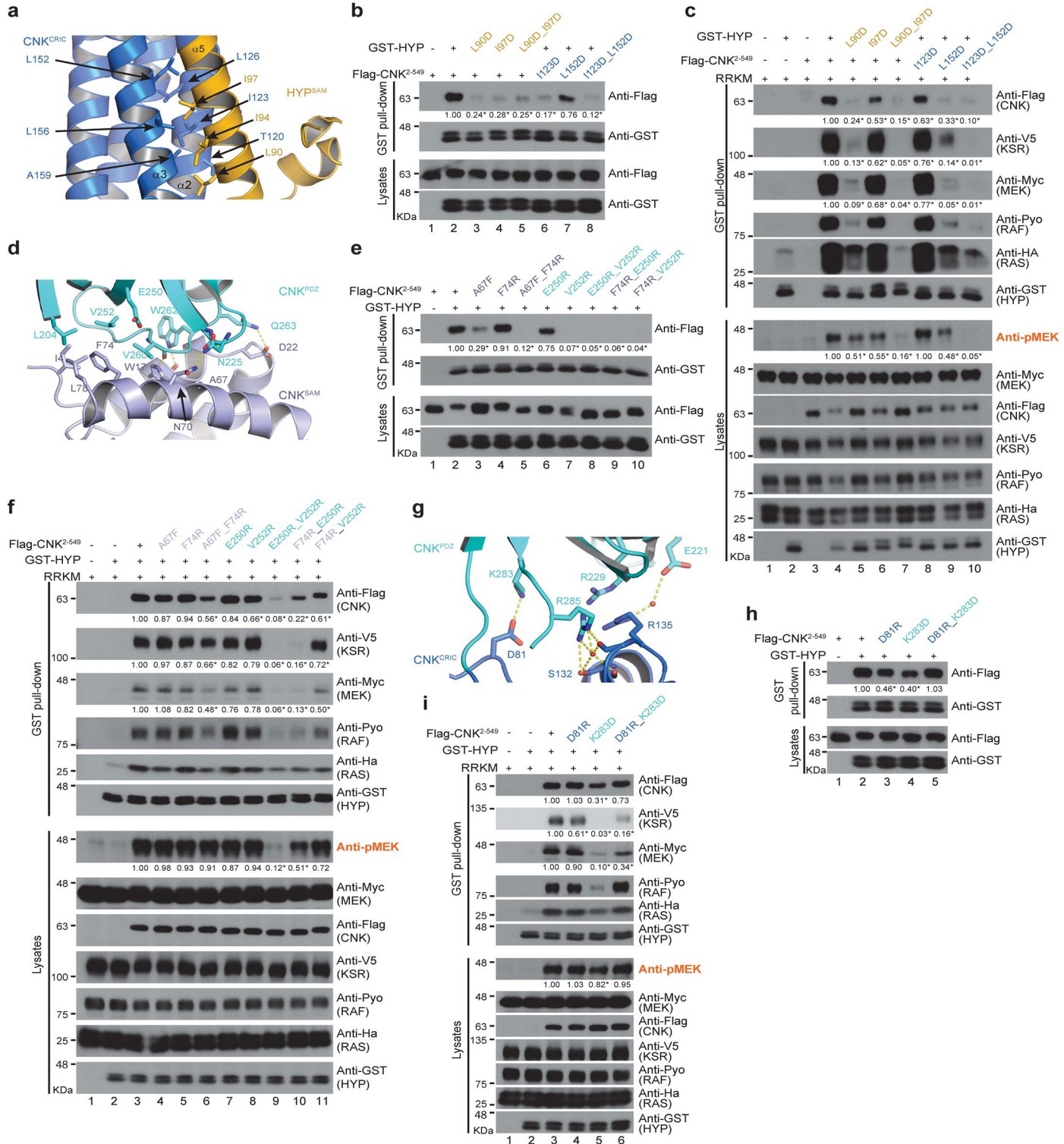

**Extended Data Fig. 2 | See next page for caption.**

**Extended Data Fig. 2 | Structural integrity of the CNK-HYP complex is required for KSR-MEK recruitment and MEK phosphorylation. a, d, g,** Ribbon representations of contacting residues of (**a**) helices α2 and α3 of CNK$^{CRIC}$ with helix α5 of HYP, (**d**) CNK$^{SAM}$ with CNK$^{PDZ}$, and (**g**) CNK$^{CRIC}$ with CNK$^{PDZ}$. **b, c, e, f, h, i,** S2 cells were transfected with the indicated plasmid constructs. Antibodies used to detect protein levels are indicated to the right of each blot. GST pull-downs were conducted on total cell lysates. Mutagenesis of the (**b, c**) CNK$^{CRIC}$-HYP, (**e, f**) CNK$^{SAM}$-CNK$^{PDZ}$, and (**h, i**) CNK$^{CRIC}$-CNK$^{PDZ}$ interacting residues generally impair the basal CNK-HYP binary interaction and reduce the ability of GST-HYP to pull-down a protein complex comprised of CNK$^{2-549}$ and RAS$^{V12}$, RAF, KSR, and MEK (denoted RRKM). In the presence of over-expressed Ha-RAS, Pyo-RAF, V5-KSR, and Myc-MEK (RRKM), only double mutations in HYP or CNK disrupt HYP interaction with CNK and with KSR-MEK and perturb RAF activation (pMEK levels). In **h** and **i**, the combination of K283D and D81R charge reversal mutations restore complex formation and signaling activity that is lost by the CNK$^{2-549}$_K283D mutation alone. Experiments in **b, c, e, f, h, i** were repeated at least three times. Signal quantifications relative to control lanes (set at 1.00) are shown under relevant blots. * denotes $P \le 0.05$ using a one-way ANOVA test. Exact P values for quantified signals in **b** are: **CNK levels:** lane 3: 0.0002; lane 4: 0.0003; lane 5: 0.0002; lane 6: <0.0001; lane 7: 0.2972; lane 8: <0.0001. Exact P values for quantified signals in **c** are: **CNK levels:** lane 5: <0.0001; lane 6: 0.0039; lane 7: <0.0001; lane 8: 0.0212; lane 9: 0.0002; lane 10: <0.0001. **KSR levels:** lane 5: <0.0001; lane 6: <0.0001; lane 7: <0.0001; lane 8: <0.0001; lane 9: <0.0001; lane 10: <0.0001. **MEK levels:** lane 5: <0.0001; lane 6: 0.0013; lane 7: <0.0001; lane 8: 0.0162; lane 9: <0.0001; lane 10: <0.0001. **pMEK levels:** lane 5: 0.0045; lane 6: 0.0092; lane 7: <0.0001; lane 8: >0.9999; lane 9: 0.003: lane 10: <0.0001. Exact P values for quantified signals in **e** are: **CNK levels:** lane 3: 0.0003; lane 4: 0.9835; lane 5: <0.0001; lane 6: 0.3252; lane 7: <0.0001; lane 8: <0.0001; lane 9: <0.0001; lane 10: <0.0001. Exact P values for quantified signals in **f** are: **CNK levels:** lane 4: 0.5998; lane 5: 0.9815; lane 6: 0.0006; lane 7: 0.3627; lane 8: 0.0072; lane 9: <0.0001; lane 10: <0.0001; lane 11: 0.0021. **KSR levels:** lane 4: 0.9979; lane 5: 0.4109; lane 6: 0.0006; lane 7: 0.1134; lane 8: 0.0525; lane 9: <0.0001; lane 10: <0.0001; lane 11: 0.0064. **MEK levels:** lane 4: 0.8782; lane 5: 0.2015; lane 6: <0.0001; lane 7: 0.0559; lane 8: 0.0844; lane 9: <0.0001; lane 10: <0.0001; lane 11: <0.0001. **pMEK levels:** lane 4: 0.9998; lane 5: 0.9972; lane 6: 0.9867; lane 7: 0.9261; lane 8: 0.9976; lane 9: <0.0001; lane 10: 0.0182; lane 11: 0.3036. Exact P values for quantified signals in **h** are: **CNK levels:** lane 3: 0.0155; lane 4: 0.0082; lane 5: 0.9943. Exact P values for quantified signals in **i** are: **CNK levels:** lane 4: 0.9913; lane 5: 0.0022; lane 6: 0.1732. **KSR levels:** lane 4: 0.0051; lane 5: <0.0001; lane 6: <0.0001. **MEK levels:** lane 4: 0.9033; lane 5: 0.0023; lane 6: 0.0135. **pMEK levels:** lane 4: 0.5193; lane 5: 0.0001; lane 6: 0.1329.

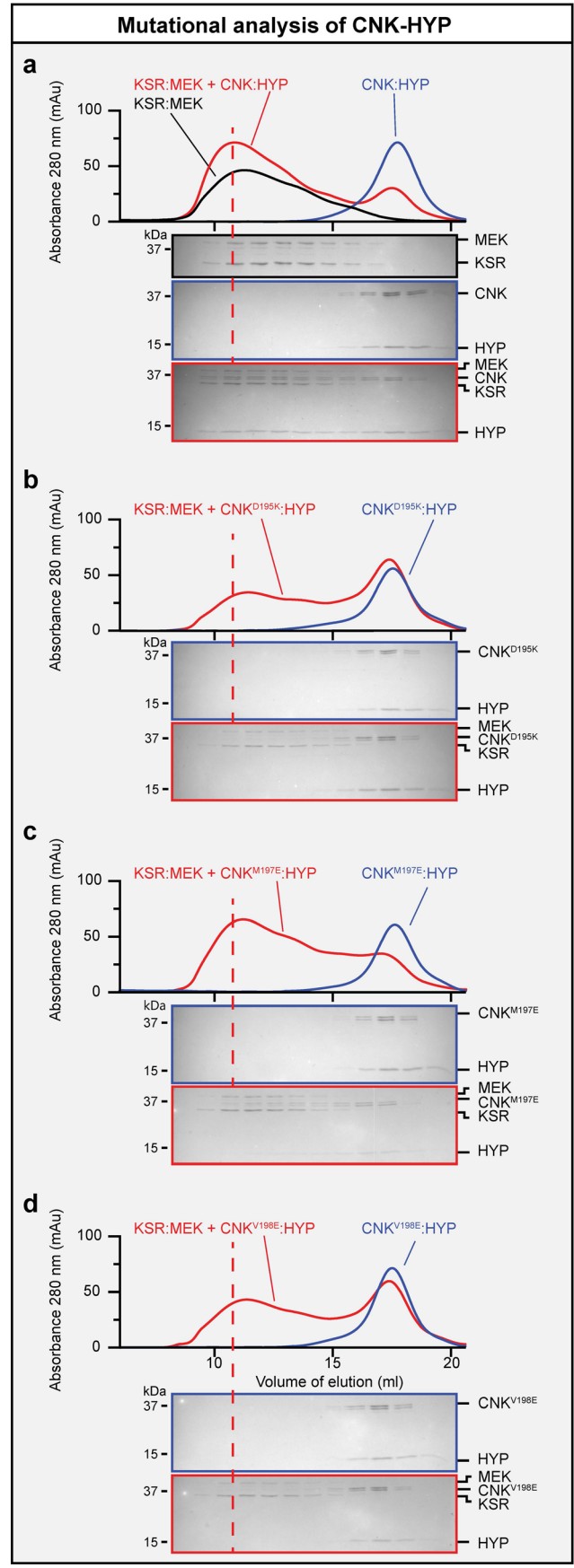

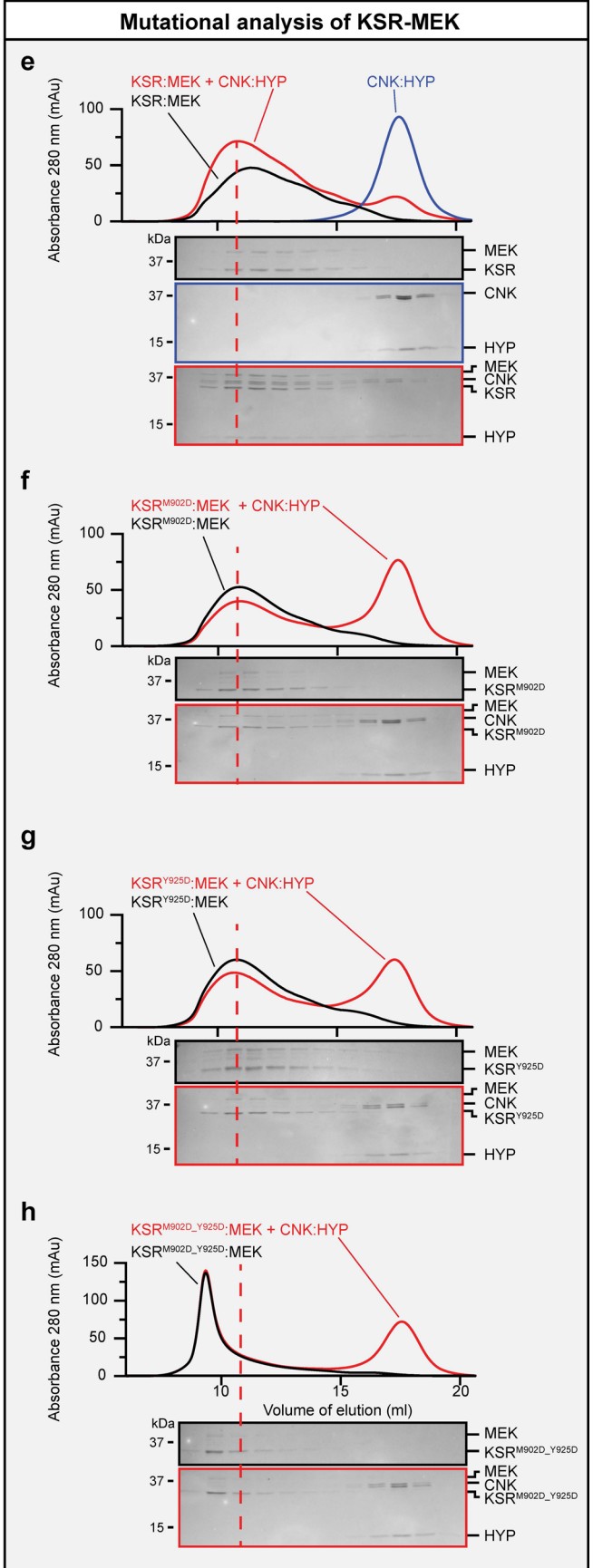

**Extended Data Fig. 3 | See next page for caption.**

**Extended Data Fig. 3 | Mutational analysis of CNK-HYP binding to KSR-MEK assessed by size exclusion chromatography. a-h**, Comparative size exclusion chromatography profiles of wild-type (WT) and mutant forms of CNK-HYP (**a-d**) and KSR-MEK (**e-h**). Black, blue and red-labeled elution profiles correspond to KSR-MEK, CNK-HYP, and KSR-MEK-CNK-HYP complexes, respectively.

Corresponding Coomassie blue-stained protein fractions resolved by SDS-PAGE are shown under each elution profile. The red vertical dashed line indicates the peak fraction of co-eluting WT KSR-MEK and CNK-HYP proteins reflecting a higher order KSR-MEK-CNK-HYP complex. All experiments were conducted in duplicate.

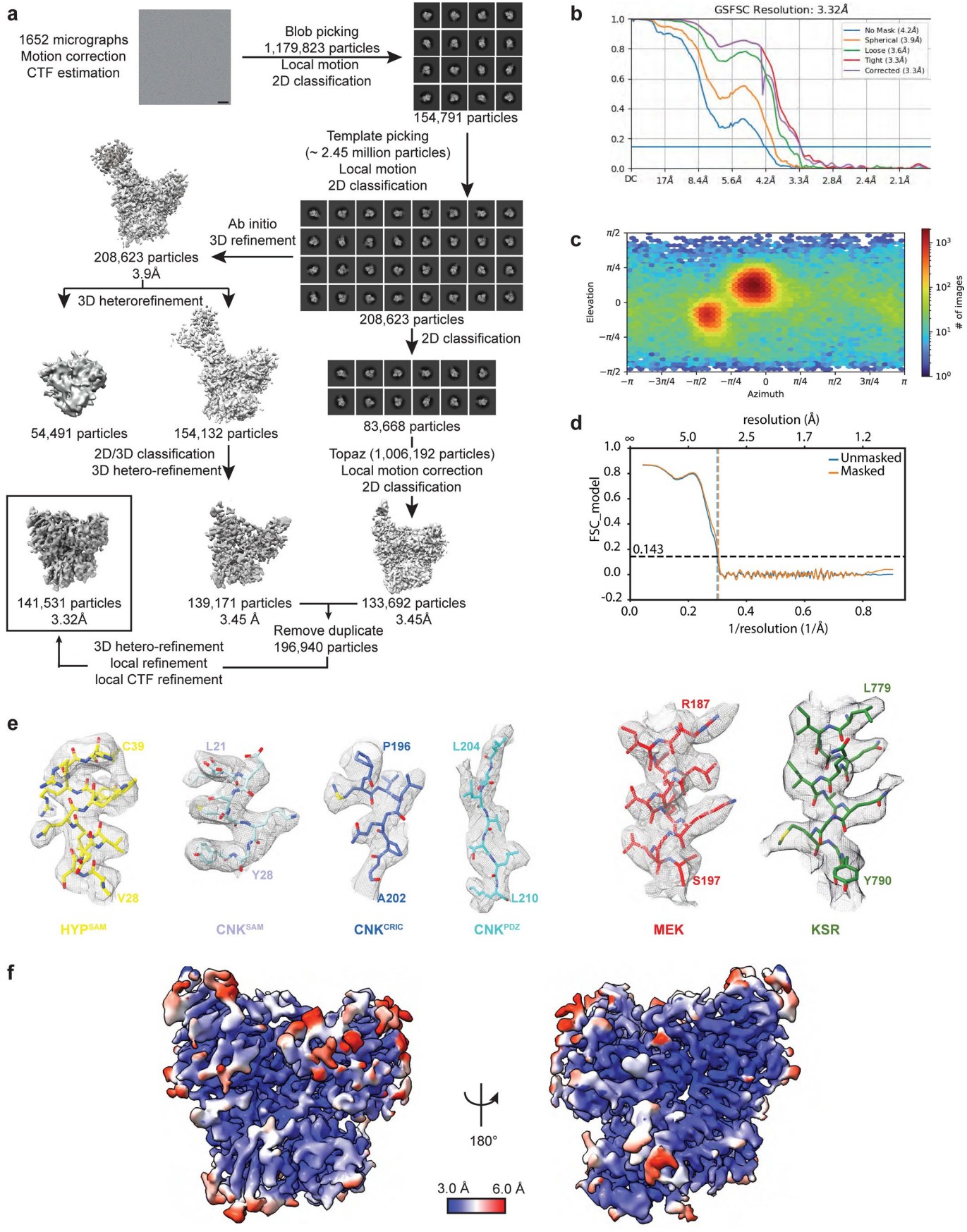

**Extended Data Fig. 4 | See next page for caption.**

**Extended Data Fig. 4 | Cryo-EM structure of a minimal complex of KSR-MEK bound to CNK-HYP. a**, Cryo-EM data processing workflow for the KSR-MEK-CNK-HYP complex performed using CryoSparc. A representative micrograph (top left, from a total of 1652 micrographs) is shown along successively refined 2D class averages and 3D models. Scale bar is 500 nm. **b**, 2D class averages from the final subset of particles used in the reconstruction. **c**, Fourier shell correlation (FSC) plots between half-maps, mask used for determination of the average resolution at FSC 0.143. **d**, Representative cryo-EM densities, corresponding to the indicated region with the fitted atomic model shown as sticks. **e**, Distribution of the particle orientations within the dataset used for 3D reconstruction. **f**, 3D representation of the KSR-MEK-CNK-HYP complex reconstruction colored according to the local resolution. **g**, Model to map Fourier shell correlation (FSC) plots calculated for the KSR-MEK-CNK-HYP complex with unmasked (blue) and masked (orange) curve. dFSC_model calculated at threshold 0.143 is indicated by the dashed line.

**a**

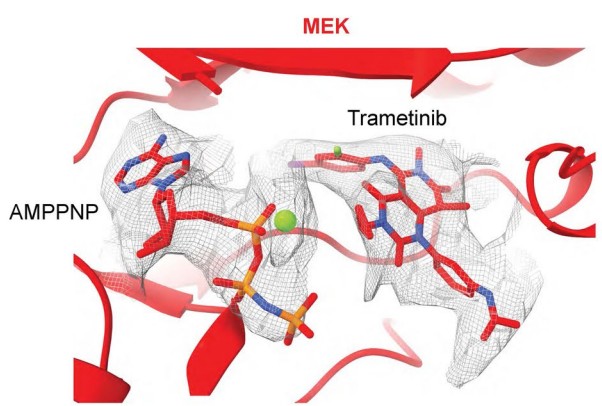

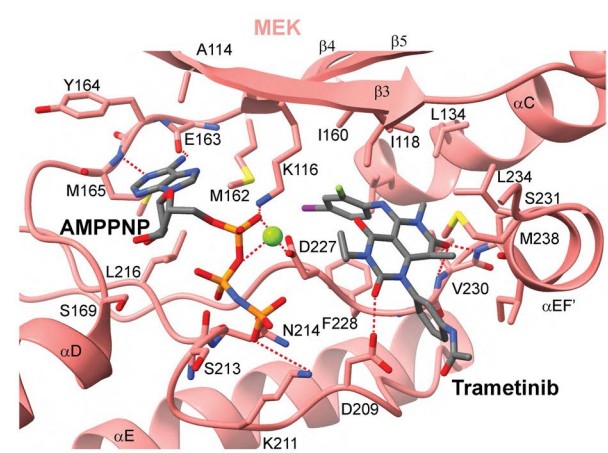

**b**

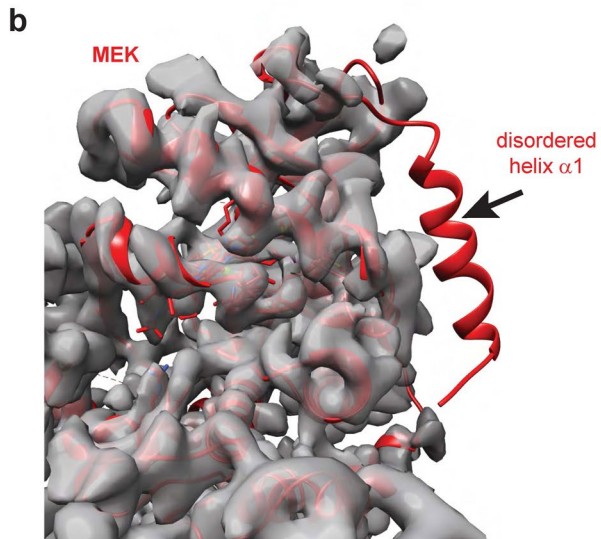

**c**

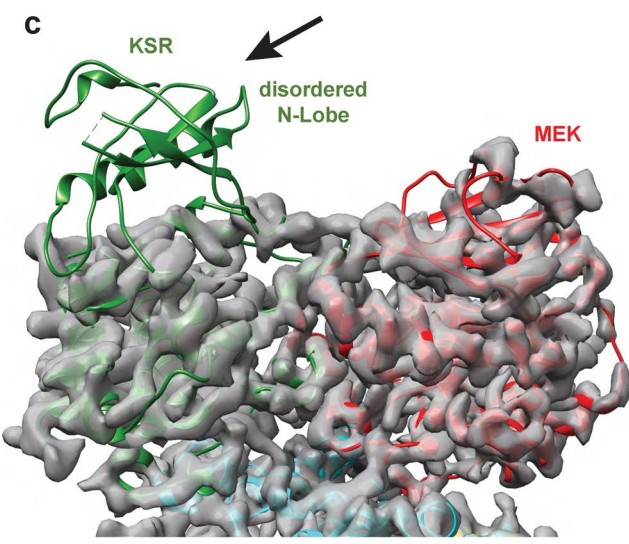

**d**

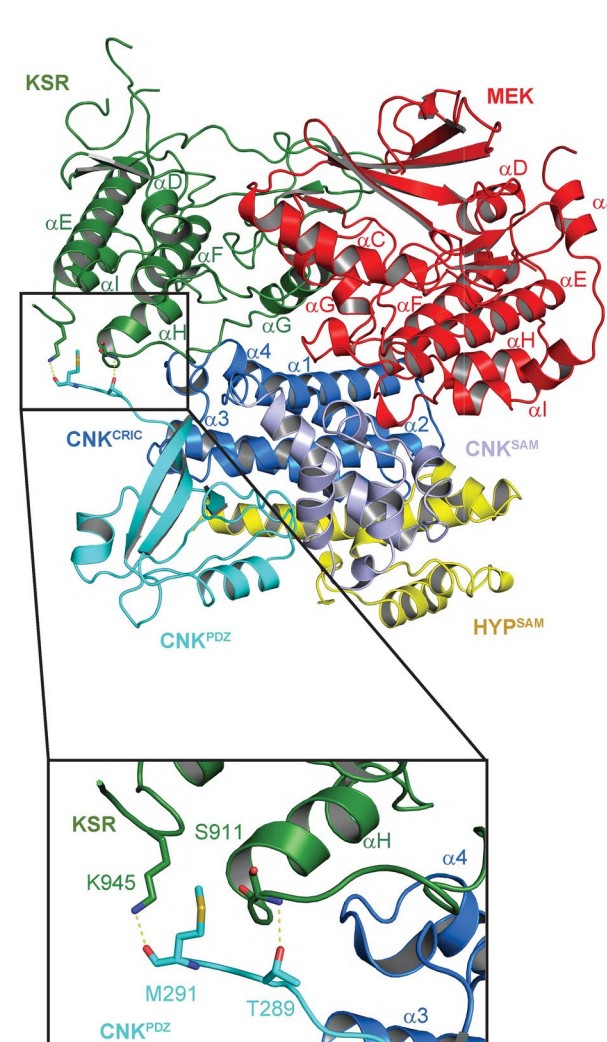

**Extended Data Fig. 5 | Analysis of the KSR-MEK-CNK-HYP structure. a**, Detailed view of the non-hydrolysable ATP-analogue AMPPNP and the MEK inhibitor trametinib bound to MEK with superimposed cryo-EM density (left panel) and with contacting residues shown as sticks (right panel). Salt and H-bonds are represented by red dashed lines. **b**, Helix α1 of MEK is not resolved in the cryo-EM map. **c**, The N-lobe of the kinase domain of KSR is not resolved in the cryo-EM map. **d**, Detailed view of the C-terminal extension of CNK^PDZ interacting with the C-lobe of the kinase domain of KSR.

**a**

KSR    MEK

αC    αEF'
αEF''
αD    αG    αF'
αG    αEF'    αC
CNK^CRIC    αEF''    αF
αEF''

HYP^SAM

CNK^SAM

CNK^PDZ

**KSR:MEK** (PDB: 7JUR)
**KSR:MEK** (in our current structure; PDB 8BW9)

**b**    KSR

RMSD=1.053Å

**c**    MEK

RMSD=0.885Å

**d**    KSR activation segment

**C-terminal portion**    **N-terminal portion**

αEF'''
P850    D817    αEF'
APEmotif    DFG motif
αEF''
G832    H837    T226
T226    A220
A220

MEK activation segment

**e**
KSR    αF 3.0Å

αE 4.8Å

αI 3.1Å
αH 3.4Å
α1
α4
CNK^CRIC

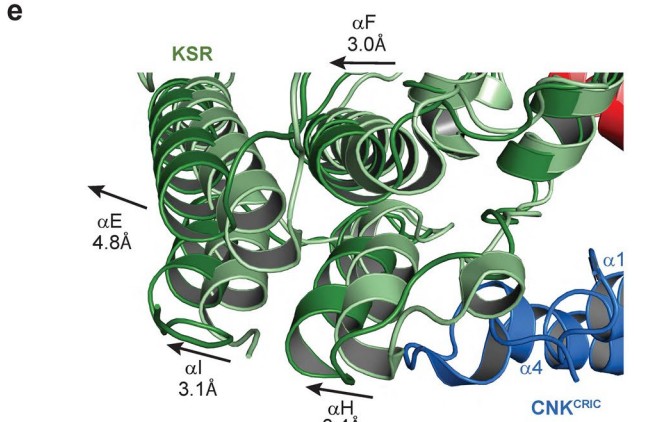

**f**
KSR
αF
site of avoided clash
L893    S895
αH    αG
D905
CNK^CRIC
P196    L908    L199    Q200    N82

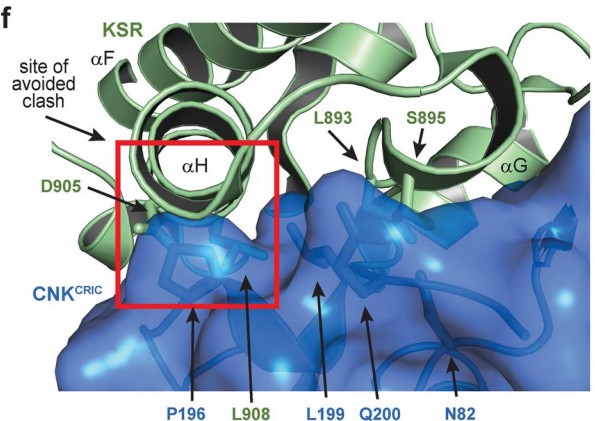

**Extended Data Fig. 6 | See next page for caption.**

**Extended Data Fig. 6 | Comparison of the KSR-MEK-CNK-HYP tetramer structure with the KSR-MEK dimer structure. a-f**, Superposition of the cryo-EM structure of the KSR-MEK-CNK-HYP complex (in dark green, dark red, blues and yellow, respectively) with the previously reported crystal structure of the KSR-MEK kinase domain complex (light green and light red, respectively; PDB ID: 7JUR ref. 19). **a**, KSR and MEK in the KSR-MEK-CNK-HYP complex, engage in a canonical face-to-face interaction centered on helices αG (inset). **b**, The C-terminal lobe of the kinase domain of KSR in the present cryo-EM structure superimposes well (RMSD of 1.053 Å) with that in the binary KSR-MEK structure (PDB ID: 7JUR). **c**, The kinase domain of MEK in the present cryo-EM structure superimposes well with that in the binary KSR-MEK structure (PDB ID: 7JUR). **d**, Cartoon representation of the activation segments of KSR (in green) and MEK (in red).**e-f**, The C-lobe of the kinase domain of KSR in the present cryo-EM structure is (**d**) shifted by ~3 Å relative to its orientation in the binary KSR-MEK structure (PDB ID: 7JUR) to avoid (**e**) a steric clash with CNK^CRIC.

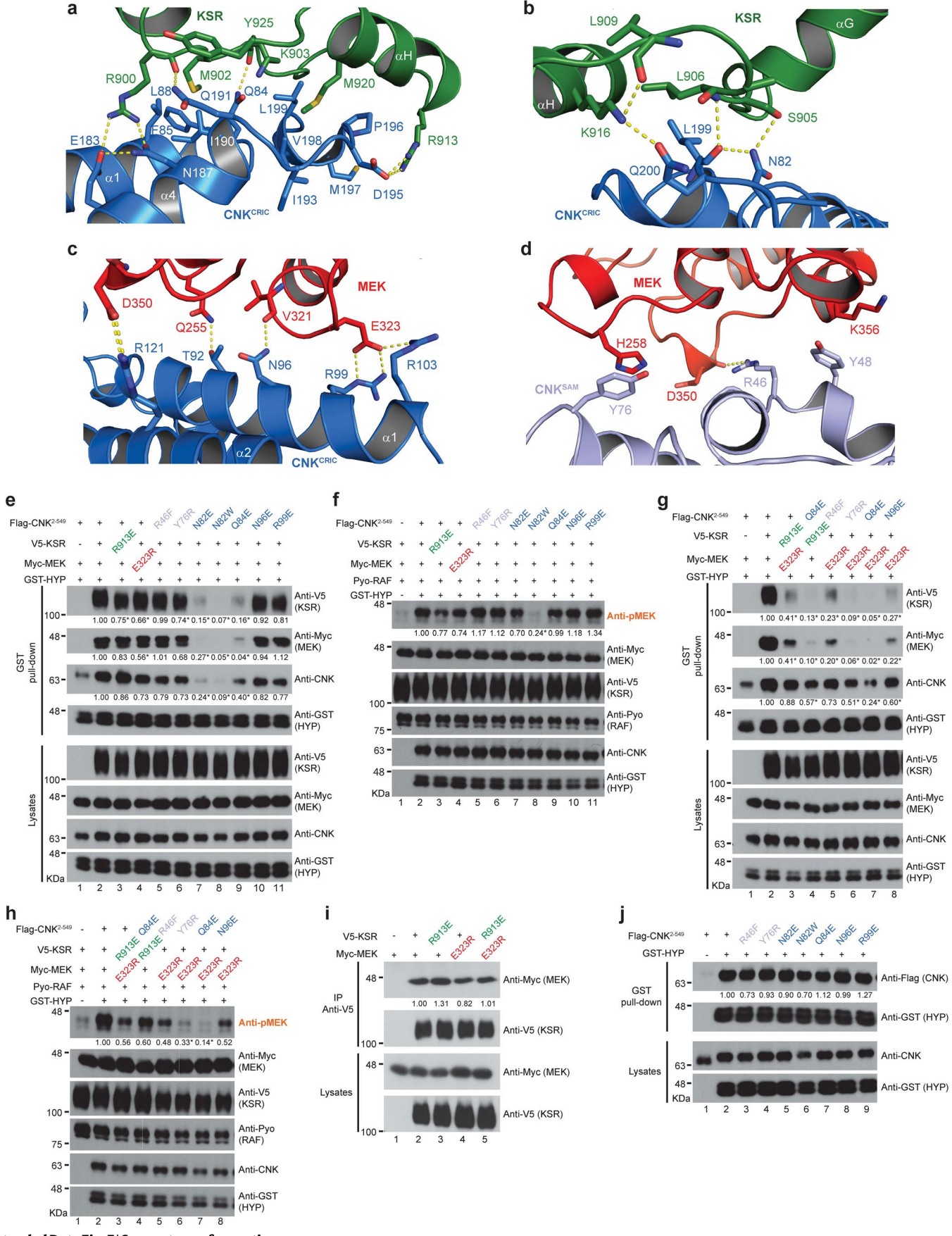

**Extended Data Fig. 7 | See next page for caption.**

**Extended Data Fig. 7 | Structure-based mutagenesis to validate KSR-MEK-CNK-HYP complex formation and function. a-d**, Ribbon representations of the binding interface between (**a,b**) KSR and CNK$^{CRIC}$ and the CNK$^{CRIC-PDZ}$ linker, between (**c**) MEK and CNK$^{CRIC}$, and between (**d**) MEK and CNK$^{SAM}$. Contact residues highlighted in stick representation. **e-j**, S2 cells were transfected with the indicated plasmid constructs. GST pull-downs (**e**, **g**, **j**) and anti-V5 immunoprecipitations (**i**) were conducted on total cell lysates. **e** and **f**, Effect of single site mutations in KSR, MEK or CNK targeting contact residues at the interface between KSR-MEK and CNK-HYP on (**e**) complex formation following GST-HYP pull-down and on (**f**) RAF activation as assessed by pMEK levels. **g-h**, Combined single point mutations of contact residues at the interface between the KSR-MEK and CNK-HYP complexes impair (**g**) higher-order complex formation as determined by GST-HYP pull-down and (**h**) RAF activation as assessed by pMEK levels. **i-j**, Single point mutations targeting contact residues between the KSR-MEK and CNK-HYP complexes do not disrupt (**i**) the binary KSR-MEK as assessed by anti-V5 co-IP or (**j**) CNK-HYP interaction as assessed by GST-HYP pull-down. Experiments in **e-j** were repeated at least three times. Signal quantifications relative to control lanes (set at 1.00) are shown under relevant blots. * denotes $P \leq 0.05$ using a one-way ANOVA test. Exact P values

for quantified signals in **e** are: **KSR levels:** lane 3: 0.0266; lane 4: 0.0013; lane 5: 0.9998; lane 6: 0.0155; lane 7: <0.0001; lane 8: <0.0001; lane 9: <0.0001; lane 10: 0.881; lane 11: 0.1292. **MEK levels:** lane 3: 0.8472; lane 4: 0.0485; lane 5: >0.9999; lane 6: 0.2486; lane 7: <0.0001; lane 8: <0.0001; lane 9: <0.0001; lane 10: 0.9995; lane 11: 0.9669. **CNK levels:** lane 3: 0.8171; lane 4: 0.159; lane 5: 0.4087; lane 6: 0.1529; lane 7: <0.0001; lane 8: <0.0001; lane 9: 0.0002: lane 10: 0.5405; lane 11: 0.2869. Exact P values for quantified signals in **f** are: **KSR levels:** lane 3: 0.5088; lane 4: 0.4075; lane 5: 0.8172; lane 6: 0.9565; lane 7: 0.2511; lane 8: 0.0003; lane 9: 0.9999; lane 10: 0.7737; lane 11: 0.1849. Exact P values for quantified signals in **g** are: **KSR levels:** lane 3: 0.0093; lane 4: 0.0003; lane 5: 0.001; lane 6: 0.0002; lane 7: 0.0001; lane 8: 0.0017. **MEK levels:** lane 3: 0.0002; lane 4: <0.0001; lane 5: <0.0001; lane 6: <0.0001; lane 7: <0.0001; lane 8: <0.0001. **CNK levels:** lane 3: 0.7591; lane 4: 0.0085; lane 5: 0.1234; lane 6: 0.0027; lane 7: <0.0001; lane 8: 0.0142. Exact P values for quantified signals in **h** are: **pMEK levels:** lane 3: 0.1076; lane 4: 0.1654; lane 5: 0.0501; lane 6: 0.01; lane 7: 0.0014; lane 8: 0.0767. Exact P values for quantified signals in **i** are: **MEK levels:** lane 3: 0.3715; lane 4: 0.7449; lane 5: 0.9999. Exact P values for quantified signals in **j** are: **CNK levels:** lane 3: 0.8649; lane 4: 0.9996; lane 5: 0.9994; lane 6: 0.8057; lane 7: 0.9976; lane 8: >0.9999; lane 9: 0.8719.

**a**

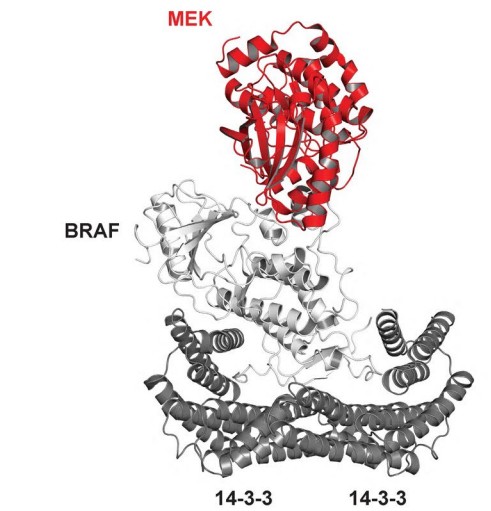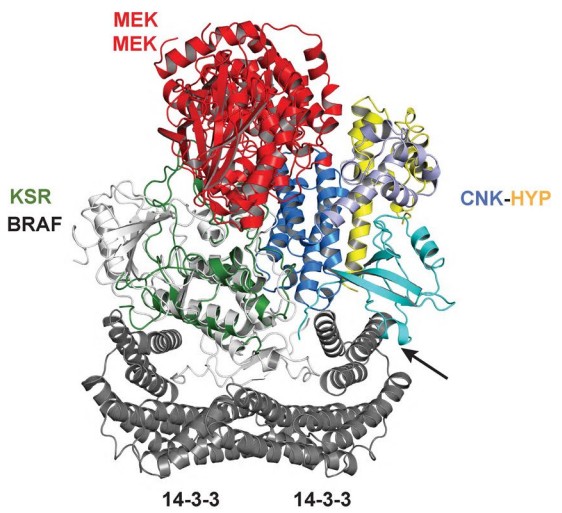

**b**

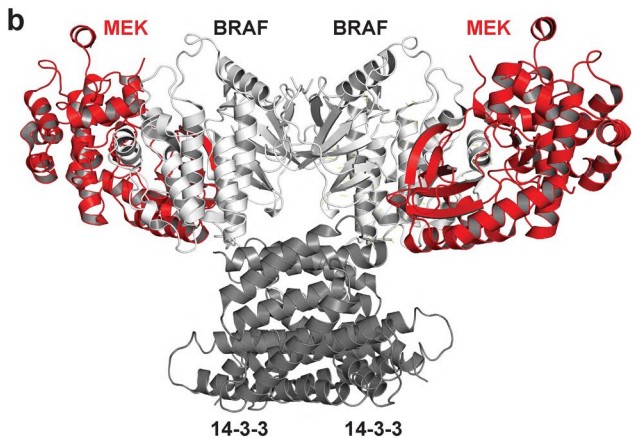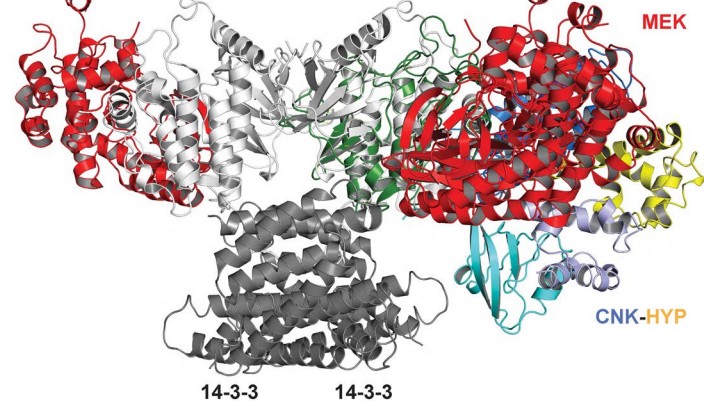

**Extended Data Fig. 8 | Binding of CNK-HYP to the kinase domain of KSR appears compatible with both the inactive and active 14-3-3-bound states of BRAF. a**, Superposition of the KSR-MEK-CNK-HYP structure on the 14-3-3-bound monomeric (auto-inhibited state) structure of BRAF (PDB ID: 6NYB). Alignment was performed using the KSR-MEK and BRAF-MEK coordinates. A small steric clash observed between the CNK$^{PDZ}$ domain and 14-3-3 is highlighted by arrow. **b**, Superposition of the KSR-MEK-CNK-HYP structure on the 14-3-3-bound dimeric (active state) structure of BRAF (PDB ID: 6Q0J[23]). Alignment was performed as in **a**. No steric clashes are detected.

# Reporting Summary

## Statistics

For all statistical analyses, confirm that the following items are present in the figure legend, table legend, main text, or Methods section.

| n/a | Confirmed | |
|---|---|---|
| ☐ | ☒ | The exact sample size (*n*) for each experimental group/condition, given as a discrete number and unit of measurement |
| ☐ | ☒ | A statement on whether measurements were taken from distinct samples or whether the same sample was measured repeatedly |
| ☐ | ☒ | The statistical test(s) used AND whether they are one- or two-sided <br> *Only common tests should be described solely by name; describe more complex techniques in the Methods section.* |
| ☒ | ☐ | A description of all covariates tested |
| ☒ | ☐ | A description of any assumptions or corrections, such as tests of normality and adjustment for multiple comparisons |
| ☒ | ☐ | A full description of the statistical parameters including central tendency (e.g. means) or other basic estimates (e.g. regression coefficient) AND variation (e.g. standard deviation) or associated estimates of uncertainty (e.g. confidence intervals) |
| ☐ | ☒ | For null hypothesis testing, the test statistic (e.g. *F*, *t*, *r*) with confidence intervals, effect sizes, degrees of freedom and *P* value noted <br> *Give P values as exact values whenever suitable.* |
| ☒ | ☐ | For Bayesian analysis, information on the choice of priors and Markov chain Monte Carlo settings |
| ☒ | ☐ | For hierarchical and complex designs, identification of the appropriate level for tests and full reporting of outcomes |
| ☒ | ☐ | Estimates of effect sizes (e.g. Cohen's *d*, Pearson's *r*), indicating how they were calculated |

*Our web collection on statistics for biologists contains articles on many of the points above.*

## Software and code

Policy information about availability of computer code

| | |
|---|---|
| Data collection | Crystallographic data were collected on the NE CAT beamline station 24-ID-C at the Advanced Photon Source (Argone, IL). <br> Cryo-EM data were collected Data on a 200 kV Talos Arctica electron microscope (Thermo Fisher Scientific), equipped with Gatan K2 summit direct electron detector (IECB, Pessac, France). <br><br> Western blot and gel band intensities were acquired using ImageJ (v. 1.54e). |
| Data analysis | Software, packages and pipelines used for crystallographic data analysis were the XDS package (v. 20170601), BLEND (v. 0.6.23), HKL2MAP (v. 0.4b-beta), SHELX (SHELXC v. 2016/1; SHELXD v. 2013/2; SHELXE v. 2018/2), PHASER (v. 2.8.1), COOT (v. 0.9), PHENIX (v. 1.20.1-4487), and MolProbity (v. v. 1.20.1-4487). <br><br> Software, packages and pipelines used for cryo-EM data analysis were SerialEM (v. 4.0.9), cryoSPARC (v. v3.2), Gctf (v. 1.06), UCSF Chimera (v. 1.15), UCSF ChimeraX (v. 1.6.1), COOT (v. 0.9), PHENIX (v. 1.20.1-4487) and ISOLDE (v. 1.4). <br><br> PyMol (v. 2.5.3) was used for structure visualization and figure preparation. <br><br> Statistical analysis was performed using GraphPad Prism (v. 9.5.1). |

For manuscripts utilizing custom algorithms or software that are central to the research but not yet described in published literature, software must be made available to editors and reviewers. We strongly encourage code deposition in a community repository (e.g. GitHub). See the Nature Portfolio guidelines for submitting code & software for further information.

## Data

Policy information about availability of data

All manuscripts must include a data availability statement. This statement should provide the following information, where applicable:
- Accession codes, unique identifiers, or web links for publicly available datasets
- A description of any restrictions on data availability
- For clinical datasets or third party data, please ensure that the statement adheres to our policy

> All data supporting the findings of the current study are available within the Article and its Supplementary Information or Source Data files. Coordinates and structures for the CNK-HYP complex and for the KSR-MEK-CNK-HYP complex have been deposited in the Protein Data Bank with accession codes 8BW8 and 8BW9, respectively. The sharpened and associated maps of the KSR-MEK-CNK-HYP complex have been deposited in the Electron Microscopy Data Bank under the accession code EMD-16281.

## Research involving human participants, their data, or biological material

Policy information about studies with human participants or human data. See also policy information about sex, gender (identity/presentation), and sexual orientation and race, ethnicity and racism.

| | |
|---|---|
| Reporting on sex and gender | N/A |
| Reporting on race, ethnicity, or other socially relevant groupings | N/A |
| Population characteristics | N/A |
| Recruitment | N/A |
| Ethics oversight | N/A |

Note that full information on the approval of the study protocol must also be provided in the manuscript.

# Field-specific reporting

Please select the one below that is the best fit for your research. If you are not sure, read the appropriate sections before making your selection.

☒ Life sciences        ☐ Behavioural & social sciences        ☐ Ecological, evolutionary & environmental sciences

For a reference copy of the document with all sections, see nature.com/documents/nr-reporting-summary-flat.pdf

# Life sciences study design

All studies must disclose on these points even when the disclosure is negative.

| | |
|---|---|
| Sample size | No statistical method was used to determine sample size. A sample size of at least n=3 was used for most experiments according to standard practice for data validation and reproducibility. Biological replicates conducted on three separate days ensured data robustness. |
| Data exclusions | No data was excluded. |
| Replication | Reproducibility was demonstrated by repeating each functional and interaction studies at least three times. |
| Randomization | This is not relevant to our study as protein interaction experiments (co-IP, western blot analysis), crystallography and cryo-EM structure determination do not require randomization. |
| Blinding | No blinding was required since randomized group allocation was not performed in this study. |

# Reporting for specific materials, systems and methods

We require information from authors about some types of materials, experimental systems and methods used in many studies. Here, indicate whether each material, system or method listed is relevant to your study. If you are not sure if a list item applies to your research, read the appropriate section before selecting a response.

## Materials & experimental systems

| n/a | Involved in the study |
|-----|-----------------------|
| ☐ | ☒ Antibodies |
| ☐ | ☒ Eukaryotic cell lines |
| ☒ | ☐ Palaeontology and archaeology |
| ☒ | ☐ Animals and other organisms |
| ☒ | ☐ Clinical data |
| ☒ | ☐ Dual use research of concern |
| ☒ | ☐ Plants |

## Methods

| n/a | Involved in the study |
|-----|-----------------------|
| ☒ | ☐ ChIP-seq |
| ☒ | ☐ Flow cytometry |
| ☒ | ☐ MRI-based neuroimaging |

## Antibodies

| | |
|---|---|
| Antibodies used | Rabbit polyclonal anti-GST (1:1000; Cell Signaling, cat. number 2622)<br>Mouse monoclonal anti-Flag, clone M2 (1:5000; Millipore Sigma-Aldrich, cat. number F3165)<br>Mouse monoclonal anti-V5, clone SV5-Pk1 (1:5000; Invitrogen, cat. number 46-0705)<br>Rabbit polyclonal anti-MEK1/2 (1:1000 Cell Signaling Technology, cat. number 9122)<br>Rabbit polyclonal anti-pMEK1/2 S217/221 (1:1000; Cell Signaling Technology, cat. number 9121)<br>Rabbit monoclonal p44/42 MAPK, clone 137F5 (1:1000; Cell Signaling, cat. number 4695)<br>Mouse monoclonal anti-pMAPK, clone MAPK-YT (1:2000; Millipore Sigma-Aldrich, cat. number M8159)<br>Mouse monoclonal anti-CNK, clone 26A6A2 (1:10), anti-Ha, clone 12CA5 (1:200), anti-Myc, clone 9E10 (1:5), and anti-Pyo (1:5) antibodies were derived from hybridoma supernatants produced in the Therrien laboratory. |
| Validation | Anti-MEK, anti-pMEK, anti-pMAPK, and anti-CNK were validated using RNA interference against their respective endogenous targets in S2 cells. Anti-GST, anti-Flag, anti-Ha, anti-Myc and anti-Pyo were validated using epitope-tagged proteins. |

## Eukaryotic cell lines

Policy information about cell lines and Sex and Gender in Research

| | |
|---|---|
| Cell line source(s) | Drosophila S2 cells were obtained from the Drosophila Genomics Resource Center (DGRC).<br>Sf9 cells were obtained from Thermo Fisher (cat. number 11496015). |
| Authentication | The S2 and Sf9 cells were not authenticated. |
| Mycoplasma contamination | The S2 cell line was regularly tested for mycoplasma.<br>The Sf9 cells were not tested for mycoplasma, but a fresh aliquot was used every 2-3 months. |
| Commonly misidentified lines<br>(See ICLAC register) | S2 and Sf9 cells were not listed in the ICLAC database. |

## Plants

| | |
|---|---|
| Seed stocks | *Report on the source of all seed stocks or other plant material used. If applicable, state the seed stock centre and catalogue number. If plant specimens were collected from the field, describe the collection location, date and sampling procedures.* |
| Novel plant genotypes | *Describe the methods by which all novel plant genotypes were produced. This includes those generated by transgenic approaches, gene editing, chemical/radiation-based mutagenesis and hybridization. For transgenic lines, describe the transformation method, the number of independent lines analyzed and the generation upon which experiments were performed. For gene-edited lines, describe the editor used, the endogenous sequence targeted for editing, the targeting guide RNA sequence (if applicable) and how the editor was applied.* |
| Authentication | *Describe any authentication procedures for each seed stock used or novel genotype generated. Describe any experiments used to assess the effect of a mutation and, where applicable, how potential secondary effects (e.g. second site T-DNA insertions, mosiacism, off-target gene editing) were examined.* |

