## [Peer Review File · Nature Structural & Molecular Biology]

Peer Review Information

Manuscript Title: The CNK-HYP scaffolding complex promotes RAF activation by enhancing KSR-MEK interaction

Corresponding author name(s): Marc Therrien, Frank Sicheri, Pierre Maisonneuve

Reviewer Comments & Decisions:

Decision Letter, initial version:

Message: 15th May 2023

Dear Dr. Therrien,

Thank you again for submitting your manuscript "The CNK-HYP scaffolding complex promotes RAF activation by enhancing KSR-MEK interaction". I apologize for the delay in responding, which resulted from the difficulty in obtaining suitable referee reports. Nevertheless, we now have comments (below) from the 3 reviewers who evaluated your paper. In light of those reports, we remain interested in your study and would like to see your response to the comments of the referees, in the form of a revised manuscript.

You will see that the reviewers appreciate the results and find the conclusions timely and of wide interest. There are, however, several concerns and suggestions that should be addressed in a revision. Specifically, we agree with reviewer #3 that a discussion exploring the mechanisms presented in the manuscript in the wider context of kinase signalling, particularly in mammals, would strengthen the manuscript. Further discussion pertaining to the conformation of the AL loops of MEK and KSR, as well as asymmetric model of Raf activation is encouraged. In line with reviewer's #2 comments, we would expect the analysis of all blots to be revisited to provide quantification. We agree with reviewer #2 that additional mutagenesis experiments to probe the KSR interaction with HYP-CNK would strengthen the conclusions.

Please be sure to address/respond to all concerns of the referees in full in a point-by-point response and highlight all changes in the revised manuscript text file. If you have comments that are intended for editors only, please include those in a separate cover letter.

We are committed to providing a fair and constructive peer-review process. Do not hesitate to contact us if there are specific requests from the reviewers that you believe are

technically impossible or unlikely to yield a meaningful outcome.

We expect to see your revised manuscript within 6 weeks. If you cannot send it within this time, please contact us to discuss an extension; we would still consider your revision, provided that no similar work has been accepted for publication at NSMB or published elsewhere.

Reporting Summary:

Please note that all key data shown in the main figures as cropped gels or blots should be presented in uncropped form, with molecular weight markers. These data can be aggregated into a single supplementary figure item. While these data can be displayed in a relatively informal style, they must refer back to the relevant figures. These data should be submitted with the final revision, as source data, prior to acceptance, but you may want to start putting it together at this point.

SOURCE DATA: we request that authors provide, in tabular form, all the data underlying the graphical representations used in figures. This is to further increase transparency in data reporting, as detailed in this editorial (<http://www.nature.com/nsmb/journal/v22/n10/full/nsmb.3110.html>).

Spreadsheets can be submitted in excel format. Only one (1) file per figure is permitted; thus, for multi-paneled figures, the source data for each panel should be clearly labeled in the Excel file; alternately the data can be provided as multiple, clearly labeled sheets in an Excel file. When submitting files, the title field should indicate which figure the source data pertains to. We request our authors to provide source data at the revision stage, so that they are part of the peer-review process. Please also include the uncropped blots and gels in the Source data file.

Data availability: this journal strongly supports public availability of data. All data used in accepted papers should be available via a public data repository, or alternatively, as Supplementary Information. If data can only be shared on request, please explain why in your Data Availability Statement, and also in the correspondence with your editor. Please note that for some data types, deposition in a public repository is mandatory - more information on our data deposition policies and available repositories can be found below: <https://www.nature.com/nature-research/editorial-policies/reporting-standards#availability-of-data>

[redacted]

Sincerely,

Katarzyna Ciazynska
(she/her)
Associate Editor
Nature Structural & Molecular Biology
<https://orcid.org/0000-0002-9899-2428>

Referee expertise:

Referee #1: signalling, structural biology

Referee #2: signalling, tumorigenesis

Referee #3: signalling, kinases, biochemistry

Reviewers' Comments:

Reviewer #1:

Remarks to the Author:

This manuscript by Maisonneuve et al reports a comprehensive analysis how the CNK and HYP proteins form a scaffolding complex and promote the KSR-MEK interaction, which in turn enhances the activity of RAF. The authors first carried out binding assays to reveal novel binding interfaces between CNK and HYP in addition to the SAM/SAM interaction known previously. They then used X-ray crystallography to solve the structure of the CNK/HYP complex, which uncovered a donut-like structure where multiple interfaces are formed between the two proteins. Surface conservation analyses helped identify residues in the CNK/HYP complex required for binding and activating the KSR/MEK complex. These findings were confirmed by a cryo-EM structure of the CNK/HYP/KSR/MEK complex, which revealed the basis by which the donut-like CNK/HYP complex binds and stabilizes the KSR/MEK complex. Based on these results and previous studies of the Raf/14-3-3 complexes, the authors provide a compelling model for the role and detailed mechanism of CNK/HYP in the activation of RAF. Malfunction of the RAF signaling pathway is a major driver of cancer, and the regulatory mechanisms of RAF are very complicated. This study reveals yet another layer of the complexity. The data in the manuscript are comprehensive and presented in a logic fashion. This paper should be published without unnecessary delay. I only have a few minor suggestions:

1. Cryo-EM densities shown in Extended figures 4d and 5a appear broken. This is caused by cutting the density too close to the atomic model. Please correct.
2. The model shown in Extended Figure 8C is a very nice summary. If there is enough space, it would be nice to put it in the last main figure. This will help readers to understand the overall model, rather than having it buried in the supplementary materials.

3. Extended Table 1. No. of reflections are very large. Are these unique reflections or total? Normally for refinement, numbers of unique reflections are reported. Please check.
4. Extended Table 2. "Electron exposure" is 1.15. This probably is the exposure per frame. The total exposure should be around 50. "Initial particle images" and "final particles images", the dots in the numbers are confusing, I suggest changing them to comma. This table also reports "model resolution" with FSC threshold 0.143, but the FSC is not shown. It is a bit surprising to see this resolution is higher (3.29) than the map resolution (3.32) with the same threshold. Please check.

Xuewu Zhang

Reviewer #2:

Remarks to the Author:

RAF activation is a complex process that has been studied many years and it is still not completely understood. This study adds one more piece to complete the puzzle. CNK was identified for the first time in 1998 by Therrien and collaborators, and ever since, mainly the Therrien group, has been elucidating its function as a multi-adaptor protein. In a previous work (PMID: 18287031), the group described the interaction between CNK and HYP recruiting KSR and, apparently, modulating RAF activation as reflected in MEK activation. The present work goes one step further and reveals the crystal structure of the multi-protein complex HYP-CNK-KSR-MEK as well as characterizes, in a structural way, its functionality in RAF activation by means of MEK activation, which is completely suitable in the scope of this journal. However, some in vivo experiments or tumor models will be required in future work to demonstrate the actual relevance of this system.

I have a few comments and suggestions that should be addressed. Comments are listed below following the order in which the figures are cited in the text:

Figure 1

Panel b: To demonstrate if MEK bound to KSR is necessary for the KSR interaction with CNK-HYP complex, an additional interesting condition could be added in figure 1b: co-expression of CNK minimal interacting region, MEK, HYP and a KSR impaired to bind MEK (W896R, as in figure 3c) or no MEK expression whatsoever.

Extended data figure 2

Main question: Is there any WT CNK or HYP in these cells that could compensate for the lack of function of the overexpressed mutants? Also, rather than only the total activated MEK in the cells, I suggest probing for pMEK in the pull downs. This is because the total pMEK in the cells may be influenced by other upstream activators or may be mediated by other scaffold proteins in the pathway.

Panel c: why is RAS pulled down with HYP even when there is no interacting CNK? And why is there no interacting MEK even in the positive control in this panel, although there is in others?

Panel f: neither CNK double mutants in lanes 9 and 10 interact with KSR, but there is still MEK activation in lane 10. What is the explanation for the difference in MEK activation between these two mutants when none of them interact with KSR?

Panel h: double mutant CNK interacting with HYP is similar to WT. Quantification would provide a better view of potential variations.

Panel i: pMEK quantification is recommended as well.

Extended data figure 3

I agree with the authors' conclusion in lines 177-178; however, the interaction upon incubation of over expressed proteins bearing domains that are able to interact does not mean that this is the case in physiological conditions in a cell system, since other proteins could be influencing or impeding these interactions. Please comment.

Figure 2

Out of all the tested mutations in CNK and HYP that could potentially impair the interaction with KSR, only M197E, in the CNKCRIC and CNKPDF linker, seems to have an impact decreasing bound KSR (fig 2c). To further analyze this, the authors performed mutagenesis of the adjacent conserved residues: D195K as well as V198E showed a decreased interaction with KSR that happened to be greater than for M197E. Even though the expression of all these mutants in the total lysates is the same, there is not the same amount of each of them interacting with HYP (Flag in figure 2d). For this reason, I would recommend quantifying the interacting KSR and normalize it to pulled down CNK for comparison among mutants.

Later in figure 2e, the authors claim that there are no changes in the interaction of these CNK mutants with HYP. However, as pointed out above, in figure 2d there is less interaction of these three mutants with HYP, which is consistent with a lower interacting KSR and, as a consequence, even to a lesser extent with MEK. On the other hand, in other figures, the presence of other components of the RAS-ERK pathway, and specifically, KSR-MEK complex seems to stabilize the CNK-HYP interaction. Another contradictory observation appears between figure 2d and 2e: the only difference between these two panels is that in figure 2e KSR and MEK are not overexpressed; nevertheless, in figure 2d there is less WT CNK interacting with HYP in the absence of KSR and MEK (lane 1), while this is not the case in the same conditions in figure 2e (lane 2).

Panel 2d: Regarding the three mutations highlighted in red in fig 2d, I would suggest testing if the effect of each of them is additive. Thus, adding four more conditions: three double mutants: M197E_D195K, D195K_V198E and M197E_V198E and a triple mutant: M197E_D195K_V198E; to determine whether there is such a combination that completely abolishes KSR interaction and perhaps CNK-HYP interaction.

Panel 2f: a shorter exposure of the Flag blot in the pull down would show, one more time, clear differences in the interacting CNK mutants (similar to 2d, opposite to 2e) again in the presence of RRKM.

Panel 2g: they use this specific double mutant to impair the interaction between CNK and KSR, however the authors did not show to what extent this mutant affects the interaction with KSR or HYP. In this figure, again, I would recommend quantification to better compare the effect in pMAPK levels between this double mutant and the lack of CNK. Another question that arises from this experiment is why was only this specific double mutant used and not other possible combinations to test their effect on MAPK activation. Specifically, why were the other two double mutants that decrease KSR recruitment by HYP-CNK separately not used in this assay?

Extended data figure 7

Panel 7f: As suggested in previous figures, pMEK levels should be quantified and compared to the control (lane 2), preferably with a lower exposure to accurately observe potential differences in pathway activation by each of these mutants and determine if the differences are consistent among the replicates.

Panel 7i: Would the co-existence of the mutation in KSR and the one in MEK disrupt their interaction, implying that this interaction somehow depends on CNK binding by both KSR

and MEK?

Panel 7j: This panel shows that there is no effect of CNK mutations on CNK-HYP interaction. However, a decreased interaction is observed in panels e and g in some cases in the presence of KSR and MEK.

Figure 4

Panel 4c: According to the authors, MEK favors RAF-KSR heterodimerization, however there is no MEK associated with the heterodimer.

Is the shift in KSR and RAF observed in lane 2 due to the lack of MEK in the complex? If that is the explanation, why is endogenous MEK not bound to the complex?

Minor:

- Typo in line 118: 296 Å should be 296 Å² if it is referring to surface area.
- Typo in line 281: extended data fig. 6h, g actually means extended data fig. 7h,g. Same case in line 282: extended data fig. 6i,j should be 7.
- Figure 3b: I assume RRM means RAS-RAF-MEK but it should be indicated in the legend.

As a general comment, while the figure legends for protein structure representations are clear and explanatory, they are not very informative when referring to the blots. The legends describe the conclusions drawn from the experiment, but it can be confusing to determine whether the assays were carried out in vitro, in which cells, etc. Although this information is sometimes mentioned in the main text and the detailed process is already explained in the materials and methods section, it would be helpful if the legends included a description of what was done rather than an interpretation of the results.

As a curiosity, why is the CNK antibody sometimes used to detect CNK (in Fig. 3 and 4), while the Flag antibody is used in other occasions (in Extended Data Fig. 2), when, in both cases, the overexpressed CNK is Flag-tagged? Is there a reason to sometimes detect the endogenous CNK as well (presumably in a separate, higher band)?

Reviewer #3:

Remarks to the Author:

This paper represents a significant contribution to our understanding of RAF activation. Specifically, it shows how a scaffold protein comprised of CNK and HYP forms a complex that helps to stabilize a KSR:MEK dimer and prime it for activation by BRAF. CNK (Connector eHancer of KSR) contains four domains – SAM, CRIK, PDZ and PH – and the authors show how the linker as well as the C-terminal domain are not required for binding to KSR:MEK. HYP (HYPhen) is a single SAM domain protein, and the two proteins form an antiparallel dimer via canonical interaction of their respective dimers. The CMK:HYP scaffold was solved by crystallography using selenomethionine while the complex of all four proteins was solved by Cryo-EM. Both models, which each show novel interfaces, were validated comprehensively with mutagenesis. Although these are *Drosophila* proteins, there are likely to be homologous interactions in the human correlates of these proteins. This is a very solid and well-written paper and should be published. There are, however, some revisions that would strengthen the paper and make it even more relevant for a general audience. There are likely some general rules here that will extend to other kinases.

Specific Comments:

1. The CNC:HYP dimer docks in a striking way to the C-lobe of both KSR and MEK and many of the same motifs from KSR and MEK are involved in this interface. The G-Helix is, for example, a prominent site of interaction in the KSR and MEK dimer and the following region up to and including the H Helix is also important for the interface as demonstrated by the mutagenesis. This builds on one of Sicheri's early paper with PRK and eIF2a where the importance of the G-Helix for substrate tethering was first demonstrated (Dar, et al, Cell, 2005). I think that this is an important point that is likely relevant for all kinases. I would suggest that the authors do an additional table where they align human and Drosophila kinase sequences of KSR, MEK, BRAF, and PRK and use PKA as a frame of reference for the elements of secondary structure which can be shown as a ribbon across the top. This would highlight the common features of the motifs and the mutation sites could be highlighted in red. I think that this would highlight the importance of these interfaces and the importance of the G-Helix as a tethering motif for protein substrates.
2. Is Ksr in an active conformation and is its activation loop phosphorylated? Since the N-lobe is disordered, I am assuming that it is not in an active conformation and that the AL is not phosphorylated but the authors should confirm this. Is the AL ordered? (See comment 4 below.)
3. In contrast to KSR, MEK does seem to be in an active conformation and its AL is extended and in an "active-like" conformation. Presumably the activation loop is not phosphorylated as this is the reaction mediated by BRAF. There are two molecules at the active site of MEK and these presumably account for the extended conformation of the AL. To what extent do ANP and the kinase inhibitor (trametinib) contribute to the stabilization of this conformation? The authors should show in more detail how these two small molecules interact at the active site and label the surrounding secondary structure elements as well as residues that interact with ANP and trametinib (Extended data Fig 5). Also AMPPNP was added to the buffer but ANP is in the structure. Did MEK hydrolyze the AMP-PNP?
4. In Extended data fig. 6a it looks as though the AL of KSR is very different in the two superimposed structures. The authors should show each structure with the motifs clearly labeled in addition to the superimposed structures. It looks as though there are significant differences. Definitely the authors should comment on the activation loops.
5. The N-lobe of KSR is clearly disordered and this is different from the KSR:MEK structure that was solved earlier by Malek. This structure has ATP in the binding site of MEK. The authors should comment further on this. It also looks as though some of the MEK N-Lobe is also disordered. The authors should also comment on this.
6. The authors should expand on the striking asymmetric model for RAF activation of MEK. This is clearly consistent with the earlier 14-3-3 structures which is very nice. Presumably only one CNK:HYP dimer could bind to the complex in a way that would allow for the other BRAF molecule to phosphorylate MEK. Is this asymmetry essential?
7. In general the supplementary information is referred to as Extended data figures and table. However. On page 9 the authors refer to Supplementary Figures 1 and 2. They should be consistent.
8. The authors should label the helices in the various figures so that the reader can get

better oriented – especially for the kinase domains. For example, the helices should be labeled in the top ribbon diagram in Figure 3a. In this top figure the mutation sites, shown beneath the ribbon diagram, should also be indicated as a black ball for the alpha carbon. This would help to orient the reader with regard to the sites and motifs and would also be important for the table suggested above where the kinases are aligned. Label helices in the Extended data Fig. 5d and also label some of the residues in Extended data Fig. 1a, 4d, and 6.

9. In the final tables of the CNK and HYP alignments the authors should just highlight with a red dot the residues that are conserved at the interfaces in their structures.

OVERALL RECOMMENDATION: This is an important paper and should definitely be published. Although there are already quite a few dense supplementary figures, I think that the above suggestions would further strengthen the paper and would highlight some of the general rules that emerge that will likely be relevant for other kinases. To what extent CNK and HYP contribute to mammalian RAF signaling remains to be determined but there will likely be some parallels with this *Drosophila* model.

Author Rebuttal to Initial comments

Reviewers' Comments:

Reviewer #1:

Remarks to the Author:

This manuscript by Maisonneuve et al reports a comprehensive analysis how the CNK and HYP proteins form a scaffolding complex and promote the KSR-MEK interaction, which in turn enhances the activity of RAF. The authors first carried out binding assays to reveal novel binding interfaces between CNK and HYP in addition to the SAM/SAM interaction known previously. They then used X-ray crystallography to solve the structure of the CNK/HYP complex, which uncovered a donut-like structure where multiple interfaces are formed between the two proteins. Surface conservation analyses helped identify residues in the CNK/HYP complex required for binding and activating the KSR/MEK complex. These findings were confirmed by a cryo-EM structure of the CNK/HYP/KSR/MEK complex, which revealed the basis by which the donut-like CNK/HYP complex binds and stabilizes the KSR/MEK complex. Based on these results and previous studies of the Raf/14-3-3 complexes, the authors provide a compelling model for the role and detailed mechanism of CNK/HYP in the activation of RAF. Malfunction of the RAF signaling pathway is a major driver of cancer, and the regulatory mechanisms of RAF are very complicated. This study reveals yet another layer of the complexity. The data in the manuscript are comprehensive and presented in a logic fashion. This paper should be published without unnecessary delay. I only have a few minor suggestions:

We thank the reviewer for the supportive assessment of our study.

1. Cryo-EM densities shown in Extended figures 4d and 5a appear broken. This is caused by cutting the density too close to the atomic model. Please correct.

We agree with the reviewer that the cryo-EM densities appear broken and have corrected the problem in the revised **Extended Data Figs. 4d** and **5a** (left panel).

2. The model shown in Extended Figure 8C is a very nice summary. If there is enough space, it would be nice to put it in the last main figure. This will help readers to understand the overall model, rather than having it buried in the supplementary materials.

As suggested by the reviewer, we have moved **Extended Data Fig. 8c** to the main figure set as **Fig. 4f**.

3. Extended Table 1. No. of reflections are very large. Are these unique reflections or total? Normally for refinement, numbers of unique reflections are reported. Please check.

The reported numbers of reflections in **Extended Data Table 1** were indeed the total number of reflections. We corrected this mistake and now report the number of unique reflections for the overall resolution (81051) and for the high-resolution shell (4475).

4. Extended Table 2. “Electron exposure” is 1.15. This probably is the exposure per frame. The total exposure should be around 50.

The “electron exposure” of 1.15 e-/Å² is indeed the exposure per frame. We have corrected this mistake and now list the total exposure as 50.53 e-/Å².

“Initial particle images” and “final particles images”, the dots in the numbers are confusing, I suggest changing them to comma.

We agree with the reviewer and have changed the dots to commas as suggested.

This table also reports “model resolution” with FSC threshold 0.143, but the FSC is not shown.

The reviewer is correct, the FSC is missing. In **Extended Data Fig. 4**, we have added a map-to-model correlation plot (as new panel g) that allows us to calculate the “model resolution” with FSC threshold 0.143. Additionally, we calculated the unmasked dFSC_model at FSC = 0.143 at 3.34 Å and report this value in the statistic table in Extended Data Table 2.

We modified the legend of **Extended Data Fig. 4** accordingly (line 857-860):

“g, Model to map Fourier shell correlation (FSC) plots calculated for the KSR-MEK-CNK-HYP complex with unmasked (blue) and masked (orange) curve. dFSC_model calculated at threshold 0.143 is indicated by the dashed line.”

It is a bit surprising to see this resolution is higher (3.29) than the map resolution (3.32) with the same threshold. Please check.

The stated reported model resolution is correct and corresponds to the dFSC_model calculated at FSC = 0.143 with the mask. Note that the dFSC_model calculated at FSC = 0.143 unmasked is 3.34Å, which is lower than the map resolution (3.32) as expected.

The observation of dFSC_model with mask being higher resolution (3.32) than the overall resolution of the map (3.29) is not uncommon and has been described previously (PMID 30198894). In addition, the discrepancy between the dFSC_model (3.29) and the dFSC_map (3.32) reported in our study is minor relative to other structures deposited to PDB (PMID 30198894).

Xuewu Zhang

Reviewer #2:

Remarks to the Author:

RAF activation is a complex process that has been studied many years and it is still not completely understood. This study adds one more piece to complete the puzzle.

CNK was identified for the first time in 1998 by Therrien and collaborators, and ever since, mainly the Therrien group, has been elucidating its function as a multi-adaptor protein. In a previous work (PMID: 18287031), the group described the interaction between CNK and HYP recruiting KSR and, apparently, modulating RAF activation as reflected in MEK activation. The present work goes one step further and reveals the crystal structure of the multi-protein complex HYP-CNK-KSR-MEK as well as characterizes, in a structural way, its functionality in RAF activation by means of MEK activation, which is completely suitable in the scope of this journal. However, some *in vivo* experiments or tumor models will be required in future work to demonstrate the actual relevance of this system.

We thank the reviewer for stating that our work is fully in line with the scope of NSMB. We agree that future work will be required to ascertain its relevance *in vivo*.

I have a few comments and suggestions that should be addressed. Comments are listed below following the order in which the figures are cited in the text:

Figure 1

Panel b: To demonstrate if MEK bound to KSR is necessary for the KSR interaction with CNK-HYP complex, an additional interesting condition could be added in figure 1b: co-expression of CNK minimal interacting region, MEK, HYP and a KSR impaired to bind MEK (W896R, as in figure 3c) or no MEK expression whatsoever.

To address this point, we have repeated the experiment in **Fig. 1b** now testing all pairwise combinations with GST-CNK²⁻⁵⁴⁹. As suggested by the reviewer, this included conditions in which the binding of KSR is assessed in the absence of MEK co-expression and vice versa. As shown in the revised **Fig. 1b**, the omission of MEK co-expression greatly impacted the binding of KSR to CNK. Likewise, the absence of KSR co-expression strongly impaired MEK binding to CNK. These results support the idea that KSR and MEK work cooperatively to bind the CNK-HYP complex. The text (**line 83-102**) has been amended to account for these new results.

Extended data figure 2

Main question: Is there any WT CNK or HYP in these cells that could compensate for the lack of function of the overexpressed mutants?

Endogenous CNK and HYP are expressed in S2 cells. However, as shown in the control lanes of **Extended Data Fig. 2**, when exogenous CNK or HYP are not expressed (e.g., lane 1 and lanes 2-3 in some instances), their contribution to interaction or activity is negligible if anything under the conditions used in these experiments.

Also, rather than only the total activated MEK in the cells, I suggest probing for pMEK in the pull downs. This is because the total pMEK in the cells may be influenced by other upstream activators or may be mediated by other scaffold proteins in the pathway.

Active (phosphorylated) MEK is typically detected in total lysates. There are two main reasons for this. First, it is not known whether pMEK associates with KSR (or any other proteins) with the same affinity as unphosphorylated MEK. For instance, the mammalian BRAF protein shows a strong loss of affinity for pMEK (Haling et al. 2014 *Cancer Cell* **26**, 402-13)). Therefore, a change of affinity for pMEK compared to the unphosphorylated form might skew the results. Secondly, as the mutations in CNK or HYP may alter (or not) KSR-MEK recruitment, which could (or not) affect pathway activity, we would not be able to untangle whether lower pMEK levels are caused by reduced binding and/or reduced pathway activity. We would like to point out that, as shown in control lanes, pMEK levels largely depend on the co-expressed Flag-CNK²⁻⁵⁴⁹ construct.

Panel c: why is RAS pulled down with HYP even when there is no interacting CNK?

We have also noticed that RAS binds weakly to GST-HYP. This is very interesting, but we have not yet investigated this observation further as it is beyond the scope of our current work. We respectfully request to leave this question for future study.

And why is there no interacting MEK even in the positive control in this panel, although there is in others?

MEK is interacting in the control lane (lane 4), but we agree with the reviewer that the MEK signal in the GST pull-down (anti-Myc panel) is overall weak for technical reasons. We have repeated this experiment and now provide a revised version of **Extended Data Fig. 2c** where MEK levels (anti-Myc) in the pulldowns are more clear.

Panel f: neither CNK double mutants in lanes 9 and 10 interact with KSR, but there is still MEK activation in lane 10. What is the explanation for the difference in MEK activation between these two mutants when none of them interact with KSR?

A longer exposure of the anti-V5 (KSR) blot (see below) shows that the F74R_E250R mutant (lane 10) interacts more strongly with GST-HYP than the E250R_V252R mutant (lane 9). Quantifications indicate a ~3-fold difference. This is also consistent with the greater CNK signal that comes down in lane 10 compared to lane 9 (see **Extended Data Fig. 2f**).

Since the comment of blot quantification has been raised by the reviewer in a number of other instances, we have decided to quantify the key signals in all blots relative to

the control lane. It should be noted that the western blot technique is semi-quantitative. It provides trends, but not absolute measurements of quantities or affinities.

Panel h: double mutant CNK interacting with HYP is similar to WT. Quantification would provide a better view of potential variations.

As suggested by the reviewer, we have quantified the CNK signals associated to GST-HYP. The data are shown in the revised **Extended Data Fig. 2h**. In this experiment (panel h), we tested separately or together the mutation of two residues predicted to interact through a salt bridge. The first residue (D81) is in the CRIC domain, whereas the second residue (K283) is in the PDZ domain. The introduced mutations reversed the original charges of these residues (i.e., D81R and K283D). When tested separately, the mutations had a ~2-fold effect on the CNK-HYP interaction (lanes 3-4). We postulated that by combining those two mutations in the same CNK molecule, a salt interaction might be reconstituted which in turn might rescue to some extent the loss of interaction. As shown in lane 5, and compared to lanes 3-4, this is indeed what we observed, namely, the double D81R_K283D CNK mutant rescues the interaction to HYP, which explains why the signal looks like WT CNK.

Panel i: pMEK quantification is recommended as well.

As suggested by the reviewer, we have now quantified pMEK levels for panel i. The results appear in the revised **Extended Data Fig. 2i** under the pMEK panel. Likewise, the double CNK mutant rescues pMEK levels compared to the K283D mutation.

Extended data figure 3

I agree with the authors' conclusion in lines 177-178; however, the interaction upon incubation of over expressed proteins bearing domains that are able to interact does not mean that this is the case in physiological conditions in a cell system, since other proteins could be influencing or impeding these interactions. Please comment.

The reviewer is correct. Our results do not prove the existence of this 4-protein complex in physiological conditions. This is why our conclusion does not make that specific statement. However, our data are fully compatible with the genetic data that place each member at the same step of the RAS-MAPK pathway, that is, between RAS and RAF (Therrien et al. *Cell* 1995, 1998, Douziech et al. *Genes Dev* 2006). Although a formal demonstration for the existence of this complex in physiological conditions will await its biochemical purification from endogenous cell lysates, we recently obtained alternative evidence leaning towards this direction. We used the miniTurbo “proteome labeling” method (Branon et al. 2018 *Nat Biotech* **36**: 880-7) in which a CNK cDNA fused to a modified BirA bacterial biotin ligase was used to identify the proximal interactome of CNK in *Drosophila* S2 cells (unpublished data). Among the top hits were KSR, MEK and HYP.

Figure 2

Out of all the tested mutations in CNK and HYP that could potentially impair the interaction with KSR, only M197E, in the CNKCRIC and CNKPDF linker, seems to have an impact decreasing bound KSR (fig 2c). To further analyze this, the authors performed mutagenesis of the adjacent conserved residues: D195K as well as V198E showed a decreased interaction with KSR that happened to be greater than for M197E. Even though the expression of all these mutants in the total lysates is the same, there is not the same amount of each of them interacting with HYP (Flag in figure 2d). For this reason, I would recommend quantifying the interacting KSR and normalize it to pulled down CNK for comparison among mutants.

There is indeed less CNK in the GST-HYP pulldowns when there is a decrease of interaction with KSR. As shown in **Fig. 1b**, the CNK-HYP interaction is stabilized by the presence of the KSR-MEK complex. The reduced CNK-HYP interaction can be also observed in the control lanes (lane 1) of **Figs. 2c and 2d**, in which the KSR and MEK constructs were not co-expressed. When the D195K or V198E CNK mutant constructs are used, the CNK-HYP interaction drops by about 4-5-fold, suggesting a strong loss of KSR (or MEK) binding to CNK-HYP. Consistent with this event, the KSR and MEK signals in the pull-downs are reduced by over 20-fold (**Figs. 2d and 2f**, lanes 5 and 7). In other words, in these conditions, the CNK-HYP interaction appears uninduced as in control lane 1.

Later in figure 2e, the authors claim that there are no changes in the interaction of these CNK mutants with HYP. However, as pointed out above, in figure 2d there is less interaction of these three mutants with HYP, which is consistent with a lower interacting KSR and, as a consequence, even to a lesser extent with MEK. On the other hand, in other figures, the presence of other components of the RAS-ERK pathway, and specifically, KSR-MEK complex seems to stabilize the CNK-HYP interaction.

The issue raised by the reviewer is analogous to their previous point. The experiment conducted in **Fig. 2e** assessed the basal CNK-HYP interaction (i.e. when KSR and MEK are not co-expressed). This is analogous to the control lane 1 in **Figs. 2c and 2d** where the GST-HYP pull-downs is conducted with the CNK²⁻⁵⁴⁹ alone (KSR and MEK are not co-expressed). The mutations indeed have a negligible effect on the binary (uninduced) CNK-HYP interaction. These results suggested that the CNK mutations are specifically impacting (orthosterically or allosterically) the interface between the CNK-HYP and KSR-MEK complexes.

Another contradictory observation appears between figure 2d and 2e: the only difference between these two panels is that in figure 2e KSR and MEK are not overexpressed; nevertheless, in figure 2d there is less WT CNK interacting with HYP in the absence of KSR and MEK (lane 1), while this is not the case in the same conditions in figure 2e (lane 2).

The conditions in **Fig. 2d**, lane 1 are identical to those in **Fig. 2e**, lane 2, which show the basal CNK-HYP interaction (not stabilized by KSR-MEK co-expression). The only difference between the CNK blots in these two experiments is their exposure time. In **Fig. 2d**, the CNK blot was exposed 10 sec., whereas the one in **Fig. 2e** was exposed 1 sec.

Panel 2d: Regarding the three mutations highlighted in red in fig 2d, I would suggest testing if the effect of each of them is additive. Thus, adding four more conditions: three double mutants: M197E_D195K, D195K_V198E and M197E_V198E and a triple mutant: M197E_D195K_V198E; to determine whether there is such a combination that completely abolishes KSR interaction and perhaps CNK-HYP interaction.

Chronologically, the mutagenesis analyses shown in **Fig. 2** (and **Fig. 3**) have been conducted prior to solving the 3D structure of the KSR-MEK-CNK-HYP complex (**Fig. 4**). They were meant to provide insights into the putative surfaces of interaction linking CNK-HYP to KSR-MEK. We reasoned that mutations in CNK that impede higher order complex formation (i.e. KSR-MEK recruitment to CNK-HYP), but not the uninduced CNK-HYP interaction, are potentially part of the surface of interaction linking CNK-HYP to KSR-MEK. Two such mutations (D195K and V198E) were identified through this analysis. They impede by more than 20-fold the recruitment of KSR (or MEK) to the CNK-HYP complex (**Fig. 2d**). In contrast, they have no to marginal effects on the uninduced CNK-HYP interaction (**Fig. 2e**). It should be noted that these mutations also severely impede the 4-protein complex formation when assessed by gel filtration but have no discernable effect on the formation of the binary CNK-HYP complex (**Extended Data Fig. 3a-d**). Now, to test whether there is an additive effect between these mutations as suggested by the reviewer, we decided to test the combination of the two strongest mutations, namely, D195K and V198E. We selected this combination for three reasons: 1- it allows to determine whether there is (or not) a greater effect on the interactions when the strongest mutations are combined; 2- it minimizes the number of permutations to test; 3- it allows to answer another point raised below

by the reviewer. Finally, to minimize the number of panels, we repeated **Figs. 2d** and **2e** and added the CNK²⁻⁵⁴⁹ D195K_V198E double mutant in these panels. As shown in the revised figures, this mutant reduced further the interaction GST-HYP with KSR and MEK (**revised Fig. 2d**), but it did not affect the uninduced CNK-HYP interaction (**revised Fig. 2e**).

Panel 2f: a shorter exposure of the Flag blot in the pull down would show, one more time, clear differences in the interacting CNK mutants (similar to 2d, opposite to 2e) again in the presence of RRKM.

These blots have now been quantified and as expected (and explained above), mutations in CNK that impair KSR-MEK binding also perturb the enhanced CNK-HYP interaction that occurs in the presence of KSR-MEK. In contrast, in conditions where KSR-MEK are not co-expressed (as in **Fig. 2e**), the basal CNK-HYP interaction is not significantly impacted by the mutations.

Panel 2g: they use this specific double mutant to impair the interaction between CNK and KSR, however the authors did not show to what extent this mutant affects the interaction with KSR or HYP. In this figure, again, I would recommend quantification to better compare the effect in pMAPK levels between this double mutant and the lack of CNK.

As requested by the reviewer, the pMAPK levels have been quantified (see revised **Fig. 2g**). The data show that, in contrast to WT CNK, the double D195K_V198E CNK mutant construct does not rescue the loss of pMAPK signal caused by depleting endogenous CNK by RNAi. Moreover, we have now tested this double mutant along the other CNK²⁻⁵⁴⁹ mutant constructs shown in **Figs. 2d** and **2e**. This double mutation impacts more strongly the interaction with KSR or MEK (**Fig. 2d**, lane 9), which also decreases the induced CNK-HYP interaction (**Fig. 2d**, lane 9). However, it does not significantly affect the basal CNK-HYP interaction (**Fig. 2e**, lane 8).

Another question that arises from this experiment is why was only this specific double mutant used and not other possible combinations to test their effect on MAPK activation. Specifically, why were the other two double mutants that decrease KSR recruitment by HYP-CNK separately not used in this assay?

We tested this specific double mutant simply because it is combining the two strongest mutations that impaired KSR-MEK recruitment to CNK-HYP.

Extended data figure 7

Panel 7f: As suggested in previous figures, pMEK levels should be quantified and compared to the control (lane 2), preferably with a lower exposure to accurately observe potential differences in pathway activation by each of these mutants and determine if the differences are consistent among the replicates.

As suggested by the reviewer, pMEK levels have been quantified (see revised **Extended Data Fig. 7f**). As reported in the text, except for the N82W mutation, the other mutations had no or marginal effects on pMEK levels. We argue that that these single mutations tested in this context still support enough complex formation leading to RAF activation. We therefore tested their effect on complex formation and pMEK levels by combining a few of them together as shown in the two subsequent panels.

Panel 7i: Would the co-existence of the mutation in KSR and the one in MEK disrupt their interaction, implying that this interaction somehow depends on CNK binding by both KSR and MEK?

We tested the suggestion made by the reviewer. As shown in the revised **Extended Data Fig. 7i** (lane 5), the co-existence of the R913E mutation in KSR and the E323R mutation in MEK did not disrupt the KSR-MEK interaction. This observation further supports the notion that the basal KSR-MEK interaction does not depend on the binding of KSR and/or MEK to the CNK-HYP complex.

Panel 7j: This panel shows that there is no effect of CNK mutations on CNK-HYP interaction. However, a decreased interaction is observed in panels e and g in some cases in the presence of KSR and MEK.

This situation is identical to the points addressed above for **Figs. 2c-f**. What is observed in panels e and g is the decrease of the enhanced CNK-HYP interaction (KSR-MEK-mediated). However, the basal (uninduced) interaction, as shown in panel j is not or marginally affected by the mutations.

Figure 4

Panel 4c: According to the authors, MEK favors RAF-KSR heterodimerization, however there is no MEK associated with the heterodimer.

In revised **Fig. 4c**, we now provide a longer exposure of the anti-Myc (MEK) blot, which shows the MEK signal in lane 3.

Is the shift in KSR and RAF observed in lane 2 due to the lack of MEK in the complex?

Yes, the co-expression of MEK leads to endogenous MAPK activation that sets off a negative feedback loop leading to RAF and KSR phosphorylation. This phenomenon has been characterized in mammalian cells (McKay et al. *PNAS* 2009 **106**: 11027-7). The mobility shifts of *Drosophila* RAF and KSR induced by MEK co-expression can be abrogated by treating cells with a MAPK inhibitor (unpublished results).

If that is the explanation, why is endogenous MEK not bound to the complex?

It should be noted that in this experiment, MEK is detected with an anti-Myc antibody, thus we did not monitor for the presence of endogenous MEK. While some endogenous MEK is likely bound to the RAF-KSR complex, the constructs used in these experiments are overexpressed and thus their respective protein levels likely surpass stoichiometrically endogenous MEK levels. This is why a MEK construct is co-expressed.

Minor:

- Typo in line 118: 296 Å should be 296 Å² if it is referring to surface area.

This has been corrected (**line 123**).

- Typo in line 281: extended data fig. 6h, g actually means extended data fig. 7h,g. Same case in line 282: extended data fig. 6i,j should be 7.

This has been corrected (**line 291-292**).

- Figure 3b: I assume RRM means RAS-RAF-MEK but it should be indicated in the legend.

This is now indicated in the legend of **Fig. 3b (line 791-792)**.

As a general comment, while the figure legends for protein structure representations are clear and explanatory, they are not very informative when referring to the blots. The legends describe the conclusions drawn from the experiment, but it can be confusing to determine whether the assays were carried out in vitro, in which cells, etc. Although this information is sometimes mentioned in the main text and the detailed process is already explained in the materials and methods section, it would be helpful if the legends included a description of what was done rather than an interpretation of the results.

As suggested by the reviewer, the legends of Figures presenting western blots have been revised to include more exhaustive experimental information.

As a curiosity, why is the CNK antibody sometimes used to detect CNK (in Fig. 3 and 4), while the Flag antibody is used in other occasions (in Extended Data Fig. 2), when, in both cases, the overexpressed CNK is Flag-tagged?

Both are interchangeable to detect the Flag-tagged CNK constructs. As these experiments were conducted over a long period of time, we happened to work with certain lots of anti-Flag antibodies that produced high background levels around 63 KDa where the Flag-CNK²⁻⁵⁴⁹ protein migrates.

In these instances, we relied on our home-made anti-CNK monoclonal antibody that we produce from a hybridoma cell line.

Is there a reason to sometimes detect the endogenous CNK as well (presumably in a separate, higher band)?

Except in **Fig. 3C** where endogenous CNK is detected in the anti-V5 (KSR) immunoprecipitates, there was no other specific reason in this study to detect endogenous CNK.

Reviewer #3:

Remarks to the Author:

This paper represents a significant contribution to our understanding of RAF activation. Specifically, it shows how a scaffold protein comprised of CNK and HYP forms a complex that helps to stabilize a KSR:MEK dimer and prime it for activation by BRAF. CNK (Connector eHancer of KSR) contains four domains – SAM, CRIK, PDZ and PH – and the authors show how the linker as well as the C-terminal domain are not required for binding to KSR:MEK. HYP (HYPhen) is a single SAM domain protein, and the two proteins form an antiparallel dimer vis canonical interaction of their respective dimers. The CMK:HYP scaffold was solved by crystallography using selenomethionine while the complex of all four proteins was solved by Cryo-EM. Both models, which each show novel interfaces, were validated comprehensively with mutagenesis. Although these are Drosophila proteins, there are likely to be homologous interactions in the human correlates of these proteins. This is a very solid and well-written paper and should be published. There are, however, some revisions that would strengthen the paper and make it even more relevant for a general audience. There are likely some general rules here that will extend to other kinases.

We thank the reviewer for their kind appreciation of our work.

Specific Comments:

1. The CNC:HYP dimer docks in a striking way to the C-lobe of both KSR and MEK and many of the same motifs from KSR and MEK are involved in this interface. The G-Helix is, for example, a prominent site of interaction in the KSR and MEK dimer and the following region up to and including the H Helix is also important for the interface as demonstrated by the mutagenesis. This builds on one of Sicheri's early paper with PRK and eIF2 α where the importance of the G-Helix for substrate tethering was first demonstrated (Dar, et al, Cell, 2005). I think that this is an important point that is likely relevant for all kinases. I would suggest that the authors do an additional table where they align human and Drosophila kinase sequences of KSR, MEK, BRAF, and PRK and use PKA as a frame of reference for the elements of secondary structure which can be shown as a ribbon across the top. This would highlight the common features of the motifs and the mutation sites could be highlighted in red. I think that this would highlight the importance of these interfaces and the importance of the G-Helix as a tethering motif for protein substrates.

As suggested by the reviewer, we generated a new figure (Supplementary Fig. 4) showing an alignment of the human and drosophila sequences of KSR, MEK, BRAF and additionally the RNA dependent protein kinase PKR and the cyclic AMP dependent protein kinase PKA, which also engage substrates (eIF2 \$\alpha\$ ) and regulatory factors/pseudo substrates (PKi) using the same infrastructure. We highlighted:

1) the key subdomains that are conserved across the protein kinase superfamily

- 2) residues in KSR and MEK that mediate interaction with CNK-HYP
- 3) the mutation sites in KSR and MEK that disrupt interaction with CNK-HYP
- 4) the residues in KSR and MEK that mediate their interaction, together with the residues in PKA and PKR that mediate contact with the pseudosubstrate PKi and the substrate eIF2 α , respectively.

We agree with the reviewer that the CNK-HYP complex strikingly docks to the same α G- α H region of both KSR and MEK, which is very close to the α G helix that is a prominent site for kinase substrate recruitment as reported previously for PKR (Dar et al. *Cell* 2005), KSR2 (Brennan et al. *Nature* 2011) and BRAF (Haling et al. *Cancer Cell* 2014).

In light of the addition of this new figure, we have added the following text to a revised discussion section (**line 385-391**):

“Interestingly, our structure shows that the CNK-HYP complex binds to the same region of KSR and MEK corresponding to the conserved helices α G and α H of the protein kinase fold (Supplementary Fig. 4). This region is close to the region that mediates substrate recruitment (in particular helix α G) by KSR, MEK²⁵, BRAF²⁶ and also by other protein kinases such as the RNA dependent protein kinase PKR²⁷ and the cyclic AMP dependent protein kinase PKA²⁸. This observation further highlights the importance of the α G- α H region as a common element of protein kinase function and control.”

2. Is Ksr in an active conformation and is its activation loop phosphorylated? Since the N-lobe is disordered, I am assuming that it is not in an active conformation and that the AL is not phosphorylated but the authors should confirm this. Is the AL ordered? (See comment 4 below.)

The reviewer is correct, in the cryo-EM map in question, we cannot make out any structural details of the entire N-lobe. However, the activation segment appears ordered and adopts an inactive conformation. Due to the limitation of resolution, we cannot confidently discern whether the activation segment is or is not in fact phosphorylated. To address this question further, we have performed trypsin digest mass spectrometry to ascertain the phosphorylation status of KSR used in our cryo-EM analysis. We only detected the non-phosphorylated form of peptides corresponding to the activation segment, suggesting that the activation segment of KSR is not phosphorylated in our cryo-EM studies. We make note of this observation in the Methods section describing the KSR-MEK protein production (please see comment #3 below).

3. In contrast to KSR, MEK does seem to be in an active conformation and its AL is extended and in an “active-like” conformation. Presumably the activation loop is not phosphorylated as this is the reaction mediated by BRAF.

In our structure, MEK is in an inactive conformation as indicated by 1) the outward conformation of the α C helix which is displaced by the presence of trametinib and 2) the outward extended conformation of the activation segment that forms an antiparallel β -sheet with the activation segment of KSR.

To determine whether the activation segment of MEK is phosphorylated, we performed a similar tryptic digest mass spectrometry analysis that was performed for KSR. Here we detected both phosphorylated and non-phosphorylated forms of peptides corresponding to the activation segment indicating that Serine 241 (equivalent to S222 in human) was partially phosphorylated in our baculovirus/insect cell protein preparations. We make note of this observation in the methods section describing the KSR-MEK protein production as follows (**line 634-640**):

“We probed the phosphorylation state of the activation segment of KSR and MEK following gel filtration purification by trypsin digest mass spectrometry. All the detected peptides of the activation segment of KSR correspond to the non-phosphorylated form suggesting that the activation segment of KSR is not phosphorylated. We detected both phosphorylated and non-phosphorylated forms of peptides corresponding to the activation segment of MEK, indicating that Serine 241 (equivalent to S222 in human) was partially phosphorylated in our baculovirus/insect cell protein preparations.”

Note however that in our cryo-EM map, we do not observe any extra density corresponding to phosphorylation of S241.

There are two molecules at the active site of MEK and these presumably account for the extended conformation of the AL

To what extent do ANP and the kinase inhibitor (trametinib) contribute to the stabilization of this conformation?

The reviewer is correct in that in our structure both AMPPNP and trametinib are bound to the active site of MEK and thus could contribute to the stabilization of the observed activation loop conformation. In particular, trametinib interacts with the N-terminal end of the activation segment – specifically the DFG motif and the 3_{10} -helix that precedes the two activation segment phosphosites – similar to what was reported previously in the structure of KSR2-MEK1 bound to trametinib (PDB 7JUR).

The authors should show in more detail how these two small molecules interact at the active site and label the surrounding secondary structure elements as well as residues that interact with ANP and trametinib (Extended data Fig 5).

As suggested by the reviewer, we modified **Extended Data Fig. 5a** to include a detailed view of the interacting residues of MEK with AMPPNP and trametinib. We also labelled the surrounding secondary structure elements.

Also AMPPNP was added to the buffer but ANP is in the structure. Did MEK hydrolyze the AMP-PNP?

The reviewer is correct, AMPPNP was added to the buffer before we prepared our cryo-EM grids and AMPPNP is also what we see in the cryo-EM structure (i.e. we see cryo-EM density for all three phosphate moieties, **Extended Data Fig. 5a**). We note that 'ANP' is the three-letter code for AMPPNP. To avoid confusion and for consistency throughout the manuscript, we have changed the figure legend of **Extended Data Fig. 5 (line 862-865)** to state AMPPNP instead of ANP. We have no evidence in our structure that MEK hydrolyzed the AMPPNP molecule bound in its active site.

4. In Extended data fig. 6a it looks as though the AL of KSR is very different in the two superimposed structures. The authors should show each structure with the motifs clearly labeled in addition to the superimposed structures. It looks as though there are significant differences. Definitely the authors should comment on the activation loops.

As suggested by the reviewer, we modified **Extended Data Fig. 6** to include a new panel (d) comparing the activation segment conformation of KSR in our structure with the activation segment conformation in the previously solved structure of KSR2-MEK1 bound to trametinib (PDB: 7JUR).

Similarities between the two structures include:

1. the activation segments of KSR and MEK form an anti-parallel β -sheet with each other.
2. the C-terminal regions extending from the anti-parallel β -sheet to the APE motif adopt similar conformations.

Differences between the two structures include:

1. in the KSR-MEK-CNK-HYP structure, the α EF' helix of KSR is not ordered and the activation loop in this N-terminal region adopts a different conformation.
2. the region connecting the APE motif to the α EF''' helix adopts a different conformation.

In light of the addition of new panel d, we amended the manuscript as follows (**line 256-261**):

“The activation segments of KSR and MEK adopt similar inactive-like conformations stabilized by an intermolecular antiparallel β -sheet as previously observed¹⁹. We note however that the activation segment of KSR in the KSR-MEK-CNK-HYP structure adopts a more unstructured conformation in comparison to the isolated KSR-MEK structure, which may be due to the disordered nature of the N-lobe of KSR in the former structure (Extended Data Fig. 6d).”

5. The N-lobe of KSR is clearly disordered and this is different from the KSR:MEK structure that was solved earlier by Malek. This structure has ATP in the binding site of MEK. The authors should comment further on this. It also looks as though some of the MEK N-Lobe is also disordered. The authors should also comment on this.

To our knowledge, the group of Shiva Malek has not published a KSR-MEK structure. The only structures of kinase domains of KSR that have been solved to date are:

- 1 structure of KSR2-MEK1 from Brennan and Dar et al. *Nature* 2011
- 1 structure of KSR2-MEK1 from Dhawan et al. *Nature* 2016
- 12 structures of KSR1:MEK1 from Khan et al. *Nature* 2020.

Presumably, the reviewer is referring to the BRAF-MEK complex structure reported by the Malek lab (Haling et al. *Cancer Cell* 2014). Comparing our new KSR-MEK structure with Malek's BRAF-MEK structure, we see nothing of great significance that has not already been reported. Therefore, we have not included further analysis.

As pointed out by the reviewer, the N-lobe of KSR is indeed disordered, which is different from the previously solved KSR-MEK structures (listed above). We believe that this may be due to the absence of ligand in the active site of the kinase domain of KSR as stated in the original manuscript (**line 252-253**):

“Disorder of the N-lobe of KSR correlated with the absence of ligand in the kinase active site.”

However, the N-lobe of MEK is not disordered, except for helix α 1 which is not visible in our cryo-EM map for reasons that remain unclear. It should be noted that a similar observation (invisible helix α 1) has been reported by the Eck lab in the cryo-EM structure of the autoinhibited BRAF:14-3-3 complex bound to MEK1 (Park et al. *Nature* 2018). Furthermore, the Malek group entirely deleted helix α 1 of the MEK1 construct used to solve the BRAF-MEK1 crystal structure (Haling et al. *Cancer Cell* 2014).

6. The authors should expand on the striking asymmetric model for RAF activation of MEK. This is clearly consistent with the earlier 14-3-3 structures which is very nice. Presumably only one

CNK:HYP dimer could bind to the complex in a way that would allow for the other BRAF molecule to phosphorylate MEK. Is this asymmetry essential?

We agree with the reviewer that presumably only one CNK-HYP complex can bind to an asymmetric KSR-RAF heterodimer, which has not previously been described. As suggested by the reviewer, we now comment on this point in our discussion as follows in **line 378-381**:

“Since RAF can exist as both symmetric homodimers and asymmetric heterodimers with KSR, the ability of CNK-HYP to drive KSR heterodimerization with RAF has the potential to influence the equilibrium between dimer configurations to fine-tune the activation of the RAS-ERK pathway.”

7. In general the supplementary information is referred to as Extended data figures and table. However. On page 9 the authors refer to Supplementary Figures 1 and 2. They should be consistent.

The **supplementary Figs. 1 and 2** correspond to the sequence alignment of HYP and CNK homologs, which are complementary to the **Fig. 2** presented in page 9. According to Nature publishing guidelines, we included the sequences alignment as Supplementary figures and not as Extended Data figures.

8. The authors should label the helices in the various figures so that the reader can get better oriented – especially for the kinase domains. For example, the helices should be labeled in the top ribbon diagram in Figure 3a. In this top figure the mutation sites, shown beneath the ribbon diagram, should also be indicated as a black ball for the alpha carbon. This would help to orient the reader with regard to the sites and motifs and would also be important for the table suggested above where the kinases are aligned. Label helices in the Extended data Fig. 5d and also label some of the residues in Extended data Fig. 1a, 4d, and 6.

As suggested by the reviewer, we labeled the helices and various residues in **Fig. 3a, Extended Data Fig. 1a, Extended Data Fig. 4d, Extended Data Fig. 5d, Extended Data Fig. 6a,e,f and Extended Data Fig. 7a-d**. In revised **Fig. 3a** we also highlighted the mutation sites as ball representation for the alpha carbon atom. We thank the reviewer for these suggestions that improve the figures and better orient the reader.

9. In the final tables of the CNK and HYP alignments the authors should just highlight with a red dot the residues that are conserved at the interfaces in their structures.

As suggested by the reviewer, in revised **Supplementary Figs. 1 and 2**, we highlighted the residues that are conserved at the interface between CNK and HYP using red triangles and the

residues that are conserved at the interface between CNK and KSR-MEK with red circles. We modified the corresponding figure legends accordingly.

OVERALL RECOMMENDATION: This is an important paper and should definitely be published. Although there are already quite a few dense supplementary figures, I think that the above suggestions would further strengthen the paper and would highlight some of the general rules that emerge that will likely be relevant for other kinases. To what extent CNK and HYP contribute to mammalian RAF signaling remains to be determined but there will likely be some parallels with this *Drosophila* model.

We thank the reviewer for their support.

To further link our findings with the function of CNK and HYP in mammals, we modified the last paragraph of the Concluding remarks as follows: **(line 396-399)**

“Interestingly, human CNK2 and the HYP homolog SAMD12 have recently been shown to control cancer cell migration by mediating ARF6 activation induced by AXL signaling³². This result indicates that minimally the CNK-HYP complex itself may provide an opportunity for targeted therapeutic intervention.”

Decision Letter, first revision:

Message: Our ref: NSMB-A47291A

14th Nov 2023

Dear Dr. Therrien,

Thank you for submitting your revised manuscript "The CNK-HYP scaffolding complex promotes RAF activation by enhancing KSR-MEK interaction" (NSMB-A47291A). It has now been seen by the original referees and their comments are below. The reviewers find that the paper has improved in revision, and therefore we'll be happy in principle to publish it in Nature Structural & Molecular Biology, pending minor revisions to satisfy the referees' final requests and to comply with our editorial and formatting guidelines.

Sincerely,
Kat

Katarzyna Ciazynska, PhD
(she/her)
Associate Editor
Nature Structural & Molecular Biology
<https://orcid.org/0000-0002-9899-2428>

Reviewer #1 (Remarks to the Author):

The authors have addressed all my questions. I have no further questions.

Reviewer #2 (Remarks to the Author):

Having carefully reviewed the revised manuscript, I am pleased to observe that all my concerns and suggestions have been thoughtfully and effectively addressed. The inclusion of suggestions from other reviewers has notably strengthened the study's robustness. In my opinion, the paper is ready for publication in NSMB.

Reviewer #3 (Remarks to the Author):

The authors did an excellent job of addressing all of the reviewer's concerns and this manuscript is highly recommended for publication.

Final Decision Letter:

Message 29th Jan 2024

:

Dear Dr. Therrien,

We are now happy to accept your revised paper "The CNK-HYP scaffolding complex promotes RAF activation by enhancing KSR-MEK interaction" for publication as an Article in Nature Structural & Molecular Biology.

As soon as your article is published, you can generate your shareable link by entering the DOI of your article here: `http://authors.springernature.com/share`. Corresponding authors will also receive an automated email with the shareable link

Your paper will be published online soon after we receive proof corrections and will appear in print in the next available issue. You can find out your date of online publication by contacting the production team shortly after sending your proof corrections.

If you have not already done so, we strongly recommend that you upload the step-by-step protocols used in this manuscript to the Protocol Exchange. Protocol Exchange is an open online resource that allows researchers to share their detailed experimental know-how. All uploaded protocols are made freely available, assigned DOIs for ease of citation and fully searchable through nature.com. Protocols can be linked to any publications in which they are used and will be linked to from your article. You can also establish a dedicated page to

collect all your lab Protocols. By uploading your Protocols to Protocol Exchange, you are enabling researchers to more readily reproduce or adapt the methodology you use, as well as increasing the visibility of your protocols and papers. Upload your Protocols at www.nature.com/protocolexchange/. Further information can be found at www.nature.com/protocolexchange/about.

Please note that *Nature Structural & Molecular Biology* is a Transformative Journal (TJ). Authors may publish their research with us through the traditional subscription access route or make their paper immediately open access through payment of an article-processing charge (APC). Authors will not be required to make a final decision about access to their article until it has been accepted. [Find out more about Transformative Journals](https://www.springernature.com/gp/open-research/transformative-journals)

Authors may need to take specific actions to achieve [compliance with funder and institutional open access mandates](https://www.springernature.com/gp/open-research/funding/policy-compliance-faqs). If your research is supported by a funder that requires immediate open access (e.g. according to [Plan S principles](https://www.springernature.com/gp/open-research/plan-s-compliance)) then you should select the gold OA route, and we will direct you to the compliant route where possible. For authors selecting the subscription publication route, the journal's standard licensing terms will need to be accepted, including [self-archiving policies](https://www.springernature.com/gp/open-research/policies/journal-policies). Those licensing terms will supersede any other terms that the author or any third party may assert apply to any version of the manuscript.

Sincerely,

Katarzyna Ciazynska, PhD
(she/her)
Associate Editor
Nature Structural & Molecular Biology
<https://orcid.org/0000-0002-9899-2428>

Click here if you would like to recommend Nature Structural & Molecular Biology to your librarian:

<http://www.nature.com/subscriptions/recommend.html#forms>